# One-Pot Syntheses of PET-Based Plasticizer and Tetramethyl Thiuram Monosulfide (TMTS) as Vulcanization Accelerator for Rubber Production

Goran Milentijević [1], Milena Milošević [2], Svetomir Milojević [1], Smiljana Marković [1], Milica Rančić [3], Aleksandar Marinković [4] and Milutin Milosavljević [1,*]

1   Faculty of Technical Science, University of Priština, Knjaza Miloša 7, 38220 Kosovska Mitrovica, Serbia
2   Institute of Chemistry, Technology and Metallurgy—National Institute of the Republic of Serbia, University of Belgrade, Njegoševa 12, 11000 Belgrade, Serbia
3   Faculty of Forestry, University of Belgrade, Kneza Višeslava 1, 11030 Beograd, Serbia
4   Faculty of Technology and Metallurgy, University of Belgrade, Karnegijeva 4, 11120 Belgrade, Serbia
*   Correspondence: milutin.milosavljevic@pr.ac.rs

**Abstract:** Styrene-butadiene (SBR) and acrylonitrile-butadiene (NBR) rubber blends with tetramethyl thiuram disulfide (TMTD) and tetramethyl thiuram monosulfide (TMTS) accelerators and environmentally friendly plasticizers, obtained from PET recycling and biobased resources (LA/PG/PET/EG/LA), were prepared. The mechanical properties of the obtained rubber products were tested and compared with those of commercial dioctyl terephthalate (DOTP). TMTS was prepared by simple and efficient one-pot synthesis from dimethylamine, carbon disulfide, potassium cyanide, and ammonium chloride as catalysts in recycled isopropanol/water azeotrope as solvent. In a comparative study, methoxide, ethoxide, iodide, and amide ions were also used. The two-step reaction mechanism of TMTS synthesis involves the oxidation of the amine salt of dimethyldithiocarbamic acid to TMTD by hydrogen peroxide and sulfur elimination from the TMTD disulfide bond. Potassium cyanide appears to be the most efficient nucleophile. The simplicity of operation, mild reaction conditions, solvent recycling, high yields, and applicability to the industrial level are the advantages of this process. Shore hardness, tensile strength, and compression test results of vulcanized blends before and after aging showed similar properties for both accelerators, while somewhat better results were obtained with LA/PG/PET/EG/LA plasticizer.

**Keywords:** tetramethyl thiuram monosulfide; tetramethyl thiuram disulfide; semi-industrial production; PET-based plasticizer; NBR and SBR rubber; rubber blends





## 1. Introduction

A new engineering approach to developing automotive tires, or the concept of green tires, adopted recently, requires continuous growth in terms of better fuel economy consumption, reduction of harmful gas emissions, improved driving safety, and increased product durability [1]. This work aimed to use styrene butadiene rubber (SBR) and butadiene-nitrile (NBR) as a network precursor in rubber blends with tetramethyl thiuram disulfide (TMTD) and tetramethyl thiuram monosulfide (TMTS) as accelerators and environmentally friendly plasticizers obtained from PET recycling to get nanocomposite materials with adequate properties [2]. The influence of synthesized accelerators and plasticizers on blend characteristics as well as the dynamic-mechanical and mechanical properties of the obtained elastomer nanocomposites will be examined and compared.

Since poly(ethylene terephthalate) (PET), a semi-crystalline, thermoplastic polyester with good mechanical properties and high transparency as a waste, represents the main ecological problem, waste PET recycling and polymer reuse have great significance for the decrease of pollution, which also reduces the plastic disposal cost [2,3]. Waste PET recycling involves four processes: re-extrusion (primary), mechanical (secondary), chemical

(tertiary), and energy recovery (quaternary). Chemical recycling of PET includes hydrolysis, alcoholysis, glycolysis, aminolysis, and other procedures applicable to producing oligomeric/monomeric products of different functionalities depending on the reagent used. In this work, PET-based eco-resins will be obtained by PET depolymerization with various poly-hydroxyl alcohols/glycols and amines. These hydroxyl/amino-terminated products (monomeric or oligomeric products, liquid to highly viscous) can be further functionalized to obtain various products, such as plasticizers for rubber blends [2,4]. In rubber production, plasticizers are generally used to help the movability of polymeric chains, enhance rubber processability, and improve its physico-mechanical characteristics. Plasticizers used in the rubber industry are mainly diesters of dibasic acids and monohydroxy alcohols, and the most common are phthalates: di(2-ethylhexyl) phthalate, better known as dioctyl phthalate (DOP), di-isononylphthalate (DINP), and a mixture of esters of aliphatic hydrophobic acids with a larger molecular weight, as well as some types of polycondensed and polymerized products and chlorinated hydrocarbons [2]. Previous research implied that di-ethylene glycol terephthalate (obtained from PET waste by alcoholic glycolysis and modified with organic acid) could act as a plasticizer with good mechanical properties [5]. Moreover, terephthalate-based plasticizers have many advantages, such as durability, material flexibility at low temperatures, and better electrical insulation.

TMTS is an active vulcanization accelerator for natural butadiene-styrene and butadiene-nitrile rubbers. It gives the cured rubber good resistance to aging and low-set compression and can be used for white and colored tires [6,7]. Thiuram accelerators represent a significant class of vulcanization accelerators due to their excellent activity since they increase vulcanization speed and processability at lower temperatures [8–10]. TMTS is also used as a fungicidally active compound [11], in the synthesis of aryl-dithiocarbamates in the reaction with aryl boronic acid in the presence of copper as a catalyst [12], and in the synthesis of diaryl sulfides (symmetric thioethers) by reaction with iodobenzenes and phenylboronic acid [13]. In previous research, TMTS was obtained by the reaction of dimethyl thiocarbamoyl chloride with the corresponding alkaline salt of dimethyldithiocarbamic acid [14]. The reaction takes place by first obtaining dimethyl thiocarbamoyl chloride [15] from TMTD by chlorination, which is isolated, and then the intermediate product reacts with sodium dimethyl dithiocarbamate according to the reaction in Scheme 1.

**Scheme 1.** Synthesis of TMTS by reaction of the sodium salt of dimethyldithiocarbamic acid with dimethylthiocarbamoyl chloride.

The synthesis of the initial tetramethyl thiuram disulfide takes place practically in two steps: in the first step, the reaction between carbon disulfide and dimethylamine occurs, and in the second step, the oxidation of the obtained dimethylamine salt of dithiocarbamic acid using an oxidizing agent occurs [16]. Most thiuram disulfides have been synthesized starting from secondary alkyl and aryl amines, carbon disulfide, sodium hydroxide [17], or ammonium hydroxide, using various oxidizing agents in the presence of catalysts and different organic solvents. The synthesis of TMTS takes place in industrial conditions by the reaction of cyanide and TMTD in an acidic environment or by the action of gaseous hydrogen cyanide on TMTD in the presence of ammonium hydroxide. TMTS is produced

in the reaction of the sodium salt of dimethyldithiocarbamic acid with phosgene [17], according to the following reaction in Scheme 2.

**Scheme 2.** Synthesis of TMTS by reaction of dimethyldithiocarbamic acid sodium salt with phosgene.

The synthesis reaction is performed by reacting a 42% solution of sodium dimethyl dithiocarbamate with phosgene, which was previously mixed with air at a molar ratio of 10:1. The temperature at which the reaction takes place is kept constant at 50 °C. The synthesis of TMTM can be performed by the reaction of halogenocyanide and alkaline dimethyl dithiocarbamate, according to the reaction Scheme 3.

**Scheme 3.** Synthesis of TMTS by reaction of halogenocyanide and alkaline dimethyl dithiocarbamate.

The reaction is carried out in an alcoholic environment at a temperature of 40 to 50 °C. Instead of cyanide bromine, it is recommended to use iodide. In the reaction of secondary amines and thiocarbamoyl halides with carbon disulfide, thiuram monosulfides are obtained, according to the reaction Scheme 4.

**Scheme 4.** Synthesis of TMTS by reaction of secondary amines and thiocarbamoyl chloride with carbon disulfide.

Wherein $R^1$ = H, a saturated, unsaturated, aliphatic, or cyclic hydrocarbon residue with 1–7 carbon atoms,

$R^2$, $R^3$, and $R^4$ = the same or different saturated or unsaturated aliphatic or cyclic hydrocarbon residues with 1–7 C-atoms.

The synthesis reaction is carried out in an organic solvent (benzene) with a slight excess of amine at 50 °C.

TMTD can be obtained by the oxidation of dithiocarbamates in high yield using $NaHCO_3$ to regulate the reaction medium's pH range from 8 to 9.5. A comparison was made using $CO_2$ as a regulator of the reaction environment, where good yields and selectivity of the reaction were achieved [18]. However, from the aspect of simplicity of the synthesis process, and especially in the industrial conditions of TMTS production, this synthesis route requires the separation of TMTD and a subsequent reaction that eliminates sulfur. Therefore, our research aims to perform a one-pot reaction using the cyanide ion and the obtained TMTD in the first reaction stage.

This manuscript presents a procedure for the one-pot synthesis of TMTS from dimethylamine, carbon disulfide, hydrogen peroxide, and the most effective nucleophile, potassium cyanide, in an isopropanol/water azeotrope (87.7–12.3%) as a solvent. The solvent mixture was recycled after synthesis. The possibilities of the reaction of eliminating sulfur from the S-S bond in TMTD using different nucleophiles ($CN^-$, $NH_2^-$, $I^-$, $MeO^-$, and $EtO^-$) in order to obtain TMTS were also examined, which is essential from the aspect that certain characteristics of the nucleophile (polarizability, basicity, nucleophilicity, etc.) can decisively affect the outcome of the reaction. The mechanism of these reactions was also studied, considering the results obtained by the isolation procedures of intermediates in TMTS syntheses. The reaction mechanism of the synthesis of TMTM by eliminating sulfur from the disulfide bond of TMTD by a cyanide ion was defined, and trial production was carried out under industrial conditions. Obtained TMTD and TMTS were used in a comparative study of the vulcanization of NBR and SBR rubber using plasticizers synthesized from waste PET glycolyzate and biobased levulinic acid (LA), i.e., LA/EG/PET/PG/LA, to perform detail analysis of their influences on material properties.

## 2. Materials and Methods

### 2.1. Materials

Dimethyl amine, carbon disulfide, hydrogen peroxide ($H_2O_2$), copper (II) sulfate, ammonium chloride, potassium hydroxide, sodium cyanide, ammonium iodide, sodium hydroxide, sodium amide, zinc chloride, xylene, tetrabutyl titanate (TBT), fructose, glycerol, $Cu(NO_3)_2 \cdot 3H_2O$, γ-alumina, silica, p-substituted phenylene diamine (PDA), antioxidant 6PPD, stearic acid, and zinc oxide were supplied by Sigma Aldrich (Munich, Germany), while potassium cyanide, sodium methoxide, sodium ethoxide, and zinc cyanide were obtained from Acros organics (Geel, Belgium). Isopropyl alcohol, benzene, dichloroethane, chloroform, absolute ethanol, sulfuric acid, methanol, hydrocloric acid, and tetrahydronaphthalene were purchased from Fisher Chemical (Hampton, UK).

Waste PET was collected from soft beverage bottles. PET bottles were crushed into small pieces (app. $0.5 \times 0.5$ cm) and washed with water and ethanol (Zorka, Šabac, Serbia) to remove impurities and residual adhesives. For the glycolysis of waste PET, i.e., depolymerization, synthesized propylene glycol—PG was used, as well as Fascat 4100 (PMC Orgnaometalli, Hoofddorp, The Netherlands) as a catalyst. All chemicals were used as received, with no further purification.

Commercial rubber and additives: Nitrile butadiene rubber (NBR) and Europrene N 3980, 28% acrylonitrile (AcN) content were supplied byVersalis (Milan, Italy); Styrene butadiene rubber Ravaflex™ SBR/1500, Ravago (Faktis-RheinChemie, Mannheim, Germany); VULKANOX MB-2/MG, the mixture of 4- and 5-methyl-2-mercaptobenzimidazole (MMBI), (Lanxess, Cologne, Germany); Carbon black Soot N—550 and Soot N—660 (Scorpio Investment Ltd., Moscow, Russia); VULKANOL FH, Xylene formaldehyde resin (Lanxess, Cologne, Germany); Ultrasil VN—3 is synthetically produced amorphous silicon dioxide (Evonik Industries, Essen, Germany); Sulfur, (Eastman Chemical Company, Rotterdam, The Netherlands); Vulkacit CBS accelerator (*N*-cyclohexyl-2-benzothiazolesulfenamide) (Lanxess, Cologne, Germany), Dioctyl terephthalate (DOTP) was supplied by BRENTAG CEE GMBH (Belgrade Serbia).

## 2.2. Synthesis of TMTS

In the experimental part, the procedure for synthesizing TMTS by one-pot oxidation of the amine salt of dimethyldithiocarbamic acid and the subsequent reaction of cyanide in the presence of an ammonium chloride catalyst were described. Also, the procedure for synthesizing TMTS by the reaction of cyanide in the presence of ammonium chloride with TMTD, obtained by oxidation of the amine salt of dimethyl dithiocarbamate with hydrogen peroxide, was described [15]. Experiments were conducted to test the possibility of TMTS synthesis by nucleophilic (thiophilic) heterolysis of the persulfide bond and elimination of sulfur from TMTD using the following nucleophiles: methoxide ($CH_3O^-$), ethoxide ($C_2H_5O^-$), iodide ($J^-$), and an amide ($NH_2^-$) ion.

### 2.2.1. Laboratory Procedure for TMTS Synthesis in One Pot Reaction

In a three-necked round bottom flask of 5000 $cm^3$, equipped with an addition funnel, thermometer, and mechanical stirrer, 1540 $cm^3$ of an azeotropic mixture of isopropyl alcohol-water (87.7–12.3%) and 437.4 $cm^3$ (4.16 mol) of a 50.0% dimethylamine solution were added, which caused the pH value to increase to 9.5. After starting the stirrer, 256.4 $cm^3$ (4.16 mol) of 98.0% carbon disulfide was added from a dropping funnel for 0.5 h while maintaining the temperature range of 28–35 °C, provided by circulating cooling water. At the end of the reaction, the pH of the reaction mixture was 6.5. At this point, 536.2 $cm^3$ of a 13.2% hydrogen peroxide solution, prepared by diluting 178.6 $cm^3$ (2.08 mol) of 35.0% hydrogen peroxide with 406.5 $cm^3$ of isopropyl alcohol/water azeotropic mixture (87.7–12.3%), was cautiously added using a dropping funnel for 0.5 h at 35–40 °C. As soon as the reaction occurred, the mixture turned yellowish due to the production of suspended TMTD particles. The end of the reaction was tested by sampling the reaction mixture, filtering it, and adding a few drops of copper (II) sulfate solution to the filtrate. In the case of the appearance of a black residue, which indicated that unreacted dithiocarbamate was still present and the reaction continued [19,20], an additional portion of hydrogen peroxide was added until a clear testing solution was obtained.

Immediately, at the end of the first step, 231.11 g (4.16 mol) of ammonium chloride and a 20% aqueous solution of potassium cyanide (4.16 mol) were added to the suspension of the obtained TMTD with intensive stirring. The reaction was carried out by adding the entire amount of potassium cyanide over two hours. Stirring was then continued for another hour while maintaining the reaction temperature at 50 °C. After the end of the reaction, the reaction mixture was filtered on a Buchner funnel (the filtrate was used for the following reaction of TMTS synthesis), the filter cake was washed with 200 $cm^3$ of water (to attain a negative SCN- test according to the method used for determination of thiocyanate ions (colorimetric method) given in the Supplementary Material), and the crystals were dried in a vacuum dryer at 60 °C (2000 Pa) until the moisture content fell below 0.5%. The raw product was recrystallized from a suitable solvent (benzene, dichloroethane, chloroform, or absolute alcohol). The yield of pure product was 96.57%, the melting temperature was 105–108 °C, and the purity was 99.2%. Purity was determined according to the literature method based on the determination of residual dithio compounds by destroying them in sulfuric acid and absorbing the produced carbon disulfide in the potassium hydroxide-alcohol solution [16].

### 2.2.2. Laboratory Procedure for TMTS Synthesis from TMTD

Laboratory experiments on the synthesis of TMTS by nucleophilic (thiophilic) heterolysis of the persulfide bond and the elimination of sulfur from TMTD [15] were presented.

### 2.2.3. Synthesis of TMTS from TMTD Using Cyanide Ions

In a 250-$cm^3$ three-necked flask equipped with a magnetic stirrer, a condenser, a thermometer, and a dropping funnel, 114 $cm^3$ of water, 14.0 g (0.057 mol) of TMTD, and 3.21 g (0.057 mol) of ammonium chloride were added. With constant stirring, a 15% aqueous sodium cyanide solution was added (3.61 g, 0.07 mol of sodium cyanide dissolved in 16 $cm^3$

of water) for two hours and stirred for another hour, providing a constant temperature of 50 °C. After adding sodium cyanide solution, the formation of yellow crystals of TMTS was observed. After the end of the reaction, the reaction mixture was filtered on a Buchner funnel, and the crystals were washed with 200 cm$^3$ of water (until a negative SCN$^-$ test) and dried in a vacuum at 60 °C (2000 Pa) until the moisture content fell below 0.5%. The raw product was recrystallized from a suitable solvent (benzene, dichloroethane, chloroform, or absolute alcohol) until a constant melting temperature was reached. The yield of yellow crystals of the synthesized TMTS was 96.6%, with a melting temperature of 105–109 °C, while data from IR, UV, and MS analysis were given in the Results section.

### 2.2.4. Laboratory Synthesis of TMTS from TMTD by Elimination of Sulfur with Methoxide Ion

In a 250-cm$^3$ three-necked flask equipped with a magnetic stirrer, a reflux condenser with a CaCl$_2$ protection tube, a thermometer, and a dropping funnel, 100 cm$^3$ of dry methanol and 1 g (0.004 mol) of TMTD were added and stirred and heated at 55 °C until complete dissolution was obtained. Then, 5 cm$^3$ (0.006 mol) of sodium methoxide solution (1.2 mol/dm$^3$) in methanol was added with intensive stirring for two hours. The reaction mixture was cooled to room temperature and divided into two parts. One portion of the reaction mixture was evaporated in a water bath to a quarter of its initial volume. The separated crystals were filtered on a Buchner funnel, washed with methanol, dried, and the melting temperature was determined. The second portion of the reaction mixture was left to separate the crystals, which were filtered on a Buchner funnel, washed with methanol, and dried, and their melting temperature, IR, and MS spectra were recorded.

Based on the obtained results, only the reactant TMTD was confirmed, meaning no reaction occurred. Results of IR, UV, and MS analyses are given in the Results section.

### 2.2.5. Laboratory Synthesis of TMTS from TMTD by Elimination of Sulfur Ethoxide Ion

In a 250-cm$^3$ three-necked flask equipped with a magnetic stirrer, a reflux condenser with a CaCl$_2$ protection tube, a thermometer, and a dropping funnel, 150 cm$^3$ of dry ethanol and 2 g (0.008 mol) of TMTD were added and heated with stirring at 60 °C to dissolve the reactant. Then 7.5 cm$^3$ (0.010 mol) of prepared sodium ethoxide solution in ethanol (13.3 mol/dm$^3$) was added with intense stirring for two hours. The reaction mixture was cooled to room temperature and divided into two parts. One part of the reaction mixture was evaporated in a water bath, whereby white crystals were separated and the melting temperature was determined. The obtained crystals were analyzed by IR and MS, and their melting temperature was determined. Based on the melting temperature and analysis, the structure of the initial reactant TMTD was confirmed, which means no reaction occurred.

The second part of the reaction mixture was evaporated under a vacuum to one-quarter of its initial volume and left to crystallize. The crystals were filtered through a Buchner funnel, washed with ethanol, and dried. Based on the results of IR and MS analysis and a specific melting temperature, the structure of the starting reactant, TMTD, was proven.

### 2.2.6. Laboratory Procedure for the Synthesis of TMTS from TMTD by Elimination of Sulfur with Amide Ion

Into a 250-cm$^3$ three-necked flask equipped with an electric heater (bullet), a magnetic stirrer, a reflux condenser with a CaCl$_2$ protection tube, a thermometer, and a dropping funnel, 150 cm$^3$ of dry xylene and 2.5 g (0.010 mol) of TMTD were added. Then, using a spatula, 1.6 g (0.020 mol) of sodium amide was carefully added, and the reaction mixture was heated at 137 °C for ten hours. After that, the reaction mixture was cooled at room temperature for 12 h, after which white crystals were separated at the bottom of the reaction flask. The crystals were separated from the xylene solution by filtration on a Buchner funnel and washed with ethanol, dried, and characterized.

The filtrate was evaporated to dryness on a vacuum evaporator and the obtained yellow crystals, which are insoluble in water, were recrystallized from methanol, dried,

and analyzed. The yield of synthesized TMTS was 50.4%, with a melting temperature of 105–108 °C, while data from IR, UV, and MS analysis were given in the Results section.

2.2.7. Synthesis of TMTS from TMTD by Elimination of Sulfur with Iodide Ion

In a 250-cm$^3$ three-necked flask equipped with an electric heater (bullet), a reflux condenser with a CaCl$_2$ protection tube, a thermometer, and a dropping funnel, 150 cm$^3$ of dry ethanol and 2.0 g (0.008 mol) of TMTD were added with moderate stirring. Then 2.0 g (0.012 mol) of ammonium iodide was added, and heating was continued at reflux for five hours. The reaction mixture turns light yellow shortly after the addition of ammonium iodide. After five hours, the reaction mixture was left for ten hours at room temperature to complete product separation. The residue was filtrated in a Buchner funnel, washed with water, dried, and characterized. The yield of yellow crystals of the synthesized TMTS was 25%, with a melting temperature of 102–106 °C, while data from IR, UV, and MS analysis were given in the Results section.

*2.3. Reaction Mechanism Study*

2.3.1. Synthesis of Sodium Dimethyl Dithiocarbamate

Into a 250-cm$^3$ three-necked flask equipped with a magnetic stirrer, reflux condenser, thermometer, dropping funnel, and water bath, 50 cm$^3$ of isopropanol/water mixture (87.7/12.3 wt%) and 33.26 cm$^3$ (0.349 mol) of 57% dimethylamine were added with vigorous stirring. The addition of 21.50 cm$^3$ (0.349 mol) of carbon disulfide (98%) was performed for one hour while maintaining a constant temperature of 30 °C. Finally, 47 cm$^3$ (0.349 mol) of 30% sodium hydroxide solution was added for one hour with continuous stirring while maintaining a temperature of 30 °C. The reaction mixture was evaporated in a vacuum evaporator until the appearance of yellow crystals. After filtration, the crystals were purified by the recrystallization of benzene. The melting point of the pure product was determined, and IR and MS analyses were performed. The yield of synthesized sodium dimethyl dithiocarbamate was 92.4%, and the melting point was 200 °C (decomposition).

2.3.2. Study of the Reaction Mechanism of TMTS Synthesis from TMTD by Sulfur Elimination with Different Nucleophiles

To define the mechanism of sulfur elimination from TMTD, experiments were conducted to isolate intermediates in the synthesis reaction and analyze the reaction products. Based on the assumption that the sulfur elimination reaction from TMTD takes place in two steps, the first step is the heterolysis of the disulfide bond by a cyanide ion. The product formed at this stage of the reaction was isolated. The second step of the reaction was the substitution of the thiocyanate anion from the resulting intermediate for the dimethyl dithiocarbamate anion, which is formed in the first step of the reaction. The intermediate dimethyl dithiocarbamate anion was isolated using zinc salt, whereby zinc bis-dimethyl dithiocarbamate (Ziram) was obtained. Determining the structure by analytical methods proved that the dimethyl dithiocarbamate anion was formed in the first step of TMTS synthesis.

2.3.3. Laboratory Procedure for the Isolation of Dimethyl Dithiocarbamate Ion Using Zn$^{2+}$ Ion in the Reaction of Synthesis of TMTS from TMTD with Cyanide Ion

Into the flask of 1000 cm$^3$ connected to the Soxhlet extractor, 600 cm$^3$ of methanol and 12.0 g (0.10 mol) of zinc cyanide were added. Afterward, 24.5 g (0.1 mol) of TMTD was introduced into the extraction tube and heated. At the same time, the solvent (methanol) started to evaporate and condense, dissolving the TMTD and returning it in the form of a solution to the flask. A partial solution concentration in the flask was observed, i.e., the separation of white crystals. The reaction mixture was filtered, the zinc dimethyl dithiocarbamate (Ziram) crystals were washed with water and dried, their melting temperature was determined, and IR and MS analyses were performed (see Section 3).

2.3.4. Laboratory Procedure for the Isolation of Dimethyl Dithiocarbamate Ion Using $Zn^{2+}$ Ion in the Synthesis Reaction of TMTS from TMTD with Amide Ion

Into the flask of 1000 cm$^3$ connected to a Soxhlet extractor, 500 cm$^3$ of xylene, 4.0 g (0.005 mol) of sodium amide, and 7.0 g (0.05 mol) of zinc chloride were added. Afterward, 12.2 g (0.05 mol) of TMTD was introduced into the extraction tube and heated. The solvent (xylene) evaporated, condensed, and dripped onto TMTD, returning to the flask as a solution. After refluxing for two hours, the reaction mixture was cooled and filtered on a Buchner funnel, and the filtrate was evaporated to dryness to give white crystals. To separate the unreacted TMTD from the possibly formed zinc dimethyl dithiocarbamate, 400 cm$^3$ of methanol was added to the resulting crystals, and the mixture was heated to start boiling. Insoluble white crystals of zinc dimethyl dithiocarbamate (Ziram) were separated by filtration, washed with water, and dried, and their melting temperature was determined and IR and MS analyses were performed (see Section 3).

2.3.5. Laboratory Procedure for the Isolation of Dimethyl Dithiocarbamate Ion Using $Zn^{2+}$ Ion in the Synthesis Reaction of TMTS from TMTD with Iodide Ion

Into the flask of 500 cm$^3$ connected to a Soxhlet extractor, 250 cm$^3$ of ethanol, 8.0 g (0.005 mol) of ammonium iodide, and 7.0 g (0.05 mol) of zinc chloride were introduced. 12.2 g (0.05 mol) of TMTD were inserted into the extraction tube, which was heated, and the solvent (ethanol) evaporated, condensed, and dripped onto the TMTD, returning to the flask as a solution. After refluxing for two hours, the reaction mixture was cooled, filtered on a Buchner funnel, and the filtrate was evaporated to dryness to give white crystals. To separate the unreacted TMTD from the possibly formed zinc dimethyl dithiocarbamate, 400 cm$^3$ of methanol was added to the resulting crystals, and the mixture was heated to boiling. Insoluble white crystals of zinc dimethyl dithiocarbamate were separated by filtration, washed with water, and dried, and their melting temperature was determined and IR and MS analysis was performed (see Section 3).

*2.4. Synthesis of LA/EG/PET/PG/LA Plasticizers*

Full details on propylene glycol (PG), levulinic acid (LA), and LA/EG/PET/PG/LA synthesis is given in Supplementary Material (Sections S2.4.1–S2.4.3).

*2.5. Rubber Blend Compounding*

Rubber blend compounding procedure is given in the Supplementary Material.

*2.6. Characterization Methods*

Structural characterization methods of the synthesized product by FTIR, $^1$H NMR, $^{13}$C NMR, MS, and UV/Vis spectroscopy are presented in the Supplementary Material, Chapter S2.6.

**3. Results**

The first series of experiments in this paper presented the definition of a laboratory procedure for the one-pot synthesis of TMTS from dimethylamine, carbon disulfide, potassium cyanide, and an ammonium chloride catalyst, using the isopropanol/water azeotrope as a solvent. The second series of experiments related to the synthesis of TMTS from TMTD using a cyanide ion as a nucleophile (thiophile). Namely, ammonium ion as an acid generates enough H$^+$ ions that the reaction takes place with a higher yield of TMTS and under mild reaction conditions compared with the usage of sulfuric acid [19]. By using ammonium chloride as a catalyst, the presence of sulfuric acid was avoided, which decomposes TMTD, as well as the possibility of accidents when working with gaseous hydrogen cyanide.

*3.1. TMTS Synthesis*

In the first step of the reaction, the amine salt of dimethyldithiocarbamic acid was formed, and its oxidation with hydrogen peroxide gave tetramethylthiuram disulfide (TMTD) in the form of a suspension in the reaction mixture. After the addition of ammonium chloride and potassium cyanide solution to the reaction mixture, sulfur was eliminated from the TMTD persulfide bond (S-S), whereby TMTS suspension was obtained. In the second step of the reaction, nucleophilic (thiophilic) heterolysis of the disulfide bond in TMTD was performed by the cyanide ion, and sulfur was eliminated from the disulfide of TMTD and TMTS was obtained with the separation of potassium thiocyanate. The resulting suspension was filtered, the cake washed with water and dried to obtain the TMTS product, and the filtrate was reused for the subsequent synthesis reaction. The presence of ammonium chloride in the reaction mixture generates $H_3O^+$ ions, so work with gaseous HCN was practically avoided, as described in the literature [19]. The described synthesis reaction procedure can be represented by a reaction Scheme 5.

**Scheme 5.** Synthesis of TMTS by heterolysis of a disulfide bond with separation of the thiocyanate ion.

The possibilities of the reaction of eliminating sulfur from the S-S bond in TMTD using different nucleophiles ($CN^-$, $MeO^-$, $EtO^-$, $NH_2^-$, and $J^-$) to obtain TMTS were examined. A study of the mechanism of these reactions is essential considering that certain characteristics of the nucleophile (polarizability, basicity, nucleophilicity, etc.) can decisively affect the outcome of the reaction.

After presenting these experimental results, the results obtained by the isolation procedures of intermediates in TMTD synthesis and the quantification of generated rodanide ions prove the reaction mechanism. To define the reaction mechanism, the results of the synthesis of the dimethyldithiocarbamic acid salt were also presented. These results were used for comparison with the analysis of the isolate intermediate (sodium salt of dimethyldithiocarbamic acid) in the first stage of the TMTD desulfurization reaction using a cyanide ion.

### 3.1.1. Results of the Synthesis of TMTS from TMTD Using Cyanide Ion

The synthesis of TMTS was optimized based on three experiments by varying the reaction conditions: reaction time, reaction temperature, and reactant concentration. The dependence of the reaction yields as a function of temperature, while other parameters remain constant: reactant concentration, mixing mode, and reaction time. This is presented in Table 1.

**Table 1.** The yield of TMTS as a function of the reaction temperature [a].

| Exp. No. | Temperature (°C) | Yield | | Melting Point [b] (°C) | Purity (%) |
|---|---|---|---|---|---|
| | | (mol) | (%) | | |
| 1 | 20 | 0.0482 | 83.49 | 103–106 | 99.1 |
| 2 | 30 | 0.0484 | 83.91 | 103–106 | 99.3 |
| 3 | 40 | 0.0490 | 84.83 | 104–106 | 99.3 |
| 4 | 50 | 0.0496 | 85.99 | 106–108 | 99.3 |
| 5 | 60 | 0.0480 | 83.25 | 104–106 | 99.2 |
| 6 | 70 | 0.0462 | 80.16 | 103–106 | 99.2 |
| 7 | 85 | 0.0463 | 80.25 | 103–106 | 99.1 |

[a] Reaction time of 2 h, quantity of reactant: TMTD 0.057 mol, NaCN 0.057 mol, $NH_4Cl$ 0.057 mol, $H_2O$ 114 cm$^3$, [b] melting point, literature data of 106–110 °C [21].

The influence of the reaction temperature was examined in the range from 20 to 85 °C, and the highest yield of 85.99% was achieved by performing experiment 4 at a temperature of 50 °C.

The dependence of the reaction product yield as a function of the reaction time while the other parameters remain constant (reactant concentration, mixing mode, and reaction temperature) are given in Table 2.

**Table 2.** The yield of TMTS as a function of reaction time [a].

| Exp. No. | Reaction Time [b] (h) | | | Yield | | Melting Point (°C) | Purity (%) |
|---|---|---|---|---|---|---|---|
| | a | b | c | (mol) | (%) | | |
| 8 | 0.5 | 0.5 | 1.0 | 0.0392 | 67.93 | 99–103 | 90.0 |
| 9 | 1.0 | 0.5 | 1.5 | 0.0501 | 86.82 | 103–107 | 98.9 |
| 10 | 1.5 | 0.5 | 2.0 | 0.0551 | 95.57 | 106–110 | 99.4 |
| 11 | 1.0 | 1.0 | 2.0 | 0.0514 | 89.16 | 104–108 | 99.0 |
| 12 | 2.0 | 1.0 | 3.0 | 0.0557 | 96.57 | 104–109 | 99.2 |

[a] Reaction temperature, 50 °C, quantity of the reactant: TMTD 0.057 mol, NaCN 0.057 mol, $NH_4Cl$ 0.057 mol, $H_2O$ 114 cm$^3$. [b] Reaction time: a—NaCN addition time, b—subsequent mixing time, c—total reaction time.

The conclusion based on the results shown in Table 2 was that the highest product yield was 96.57%, achieved in a reaction time of 3 h with the addition of NaCN for two hours and subsequent mixing for another hour. The yield dependence of the reaction product is disclosed as a function of the concentrations of the reactants. At the same time, the other parameters remained constant: reaction time, mixing mode, and reaction temperature, as given in Table 3.

The results shown in Table 3 showed that the highest product yield was 97.01%, achieved in experiment 14, where a slight excess of sodium cyanide was used. Using an excess of TMTD (experiment 16) resulted in a lower product yield. Most likely, the unreacted TMTD in the reaction mixture affected the efficient separation of TMTS by recrystallization from methanol.

In all the mentioned experiments, water was used in the amount of 114 cm$^3$, whereby optimal yields of reaction products were achieved. Reduction of the amount of water as a solvent did not give satisfactory yields due to inefficient suspension mixing. Thus, a lower degree of conversion was obtained. Also, based on the results shown in Tables 1–3, it can be seen that the melting temperatures of the obtained TMTS products were appropriate in comparison with the literature data [21] under the optimal synthesis reaction conditions.

**Table 3.** The yield of TMTS as a function of the amounts of reactants [a].

| Exp. No | Initial Reactant Amount (mol) | | | Yield | | Melting Point (°C) | Purity (%) |
|---|---|---|---|---|---|---|---|
| | TMTD | NaCN | NH$_4$Cl | (mol) | (%) | | |
| 13 | 0.057 | 0.067 | 0.057 | 0.0552 | 96.84 | 105–109 | 99.2 |
| 14 | 0.057 | 0.070 | 0.057 | 0.0553 | 97.01 | 106–109 | 99.1 |
| 15 | 0.067 | 0.057 | 0.057 | 0.0519 | 91.05 | 103–105 | 90.1 |
| 16 | 0.070 | 0.057 | 0.057 | 0.0490 | 85.96 | 101–104 | 90.0 |
| 17 | 0.057 | 0.057 | 0.067 | 0.0551 | 96.66 | 105–109 | 99.2 |
| 18 | 0.057 | 0.057 | 0.070 | 0.0552 | 96.84 | 105–109 | 99.2 |

[a] Reaction temperature: 50 °C; reaction time: 3 h.

3.1.2. Results of the Synthesis of TMTS from TMTD Using the Amide Ion

As described in the Experimental part, the synthesis of TMTS from TMTD by an amide ion was performed by varying the reaction parameters: reactant concentration, reaction time, reaction temperature, and reaction medium (xylene, tetrahydronaphthalene). An overview of the obtained experimental results is presented in Table 4.

**Table 4.** Results of TMTS synthesis using an amide ion.

| Exp. No. | Reaction Conditions | | | | | Yield (%) | Melting Point, (°C) | Purity (%) |
|---|---|---|---|---|---|---|---|---|
| | Initial Reactant Amounts (mol) | | Solvent [a] (cm$^3$) | T (°C) | Time (h) | | | |
| | TMTD | NaNH$_2$ | | | | | | |
| 19 | 0.010 | 0.010 | 110 | 60 | 5.0 | - | - | - |
| 20 | 0.010 | 0.020 | 120 | 60 | 7.0 | - | - | - |
| 21 | 0.010 | 0.020 | 150 | 136 | 10.0 | 50.4 | 106–109 | 99.3 |
| 22 | 0.010 | 0.020 | 120 | 120 | 7.0 | 34.6 | 106–109 | 99.2 |

[a] Xylene was used as a solvent in experiments 19–21 and tetrahydronaphthalene in experiment 22 t.

The results shown in Table 4 implied that in experiment 19, the reaction temperature was 60 °C and the ratio of reactants was equimolar. In experiment 20, amide was used in excess, and the reaction time was extended to seven hours. By analyzing the reaction mixture, it was determined that no TMTS synthesis reaction occurred in both experiments. In experiment 21 (Experimental part 2.2.4), a TMTS product yield of 50.4% was achieved. This reaction took place at elevated temperatures with reflux, a prolonged reaction time, and excess amide. The white crystals, which were separated by filtration of the reaction mixture, had a melting point of 149–152 °C, and IR and MS images indicated that they were unreacted TMTD.

Evaporation of the filtrate yields yellowish crystals after recrystallization from methanol, and analysis of the IR and MS spectra and a melting temperature of 106–109 °C confirm that the yellow crystals represent the obtained TMTS product (Figures S4 and S5).

Experiment number 22 (Table 4) showed that an increase in the reaction temperature had not resulted in a higher conversion degree (the yield of TMTS was 34.6%). Namely, the obtained TMTS and unreacted TMTD, dimethylthiourea, and sodium dimethyl dithiocarbamate were determined by analyzing the reaction mixture. This fact suggests that the reaction is partially performed in the first step, yielding dimethyl dithiocarbamate and dimethyl thiosulfenamide. By further heating the reaction mixture, in addition to the formation of TMTS, thiourea was isolated by eliminating sulfur from dimethyl thiosulfenamide and the decomposition of the present TMTD. These undesirable reactions lead to a lower conversion degree of reactants to products and the separation of by-products.

### 3.1.3. Results of the Synthesis of TMTS from TMTD Using Iodide Ion

The described experiments in this paper present different experiments on the synthesis of TMTS by eliminating sulfur from the S-S bond in TMTD with an iodide ion. The synthesis reactions were carried out by varying the parameters: reactant concentration, reaction time, reaction temperature, and solvent. An overview of the obtained experimental results is presented in Table 5.

**Table 5.** Results of TMTS synthesis using the iodide ion.

| Exp. No. | Reaction Conditions | | | | | | Yield (%) | Melting Point, (°C) | Purity (%) |
|---|---|---|---|---|---|---|---|---|---|
| | Initial Reactant Amount (mol) | | | Solvent [a] (cm$^3$) | T (°C) | Time (h) | | | |
| | TMTD | NH$_4$I | KI | | | | | | |
| 23 | 0.008 | 0.014 | - | 120 | 60 | 6.0 | - | - | - |
| 24 | 0.008 | 0.014 | - | 150 | 105 | 6.0 | - | - | - |
| 25 [b] | 0.008 | 0.014 | - | 150 | 70 | 6.0 | - | - | - |
| 26 | 0.008 | - | 0.008 | 150 | 70 | 6.0 | - | - | - |
| 27 | 0.008 | 0.012 | - | 150 | 80 | 5.0 | 25.0 | 106–110 | 99.4 |

[a] The solvent used in experiments 23–26 was water, and in experiment 27 it was absolute ethanol. [b] In experiment 25, acetic acid was added to adjust pH 4.

The results shown in Table 5 indicated that a yield of 25% was achieved in experiment 27, in which an excess of reactant ammonium iodide was used and ethanol as the reaction medium. Numerous attempts to prolong the reaction time and increase the concentration of reactants did not yield a higher yield. In the other experiments, the conversion of reactants into products was not achieved.

Based on the obtained results, it can be noticed that the highest product yield was achieved with the use of cyanide ions. The use of amide and iodide resulted in lower yields, while the synthesis of TMTS did not occur with methoxide and ethoxide reactants under the tested conditions.

The cyanide ion was the most reactive nucleophile in the substitution reaction on the S-atom of the S-S bond of TMTD. Its reactivity was also high in $S_N2$ reactions on the C-atom of methyl iodide. The cyanide ion has a basic reaction, as it is of great importance because of its strong nucleophilicity, which was reflected in the first step of the synthesis reaction where it replaced the dimethyl dithiocarbamate anion, which is also a very strong nucleophile.

The amide ion ($NH_2^-$) ion is also basic, but it is a less reactive nucleophile than the $CN^-$ ion in the substitution reaction at the S-atom of TMTD. It was used in these reactions in organic solvents (xylene, tetrahydronaphthalene).

Iodide ions ($I^-$) were used in the substitution reaction on the S-atom of TMTD in ethanol as a solvent. It was shown to be less efficient than the amide ion for this reaction. Even though the iodide ion is a strong nucleophile in $S_N2$ reactions on the C-atom and the solvation effect is not very significant for the reactions of the iodide ion, its reactivity to the S-S bond of TMTD was weaker than that of the cyanide and amide ions.

Methoxide and ethoxide ions in the reaction of sulfur elimination from TMTD did not give the product of the TMTS reaction, regardless of the fact that they are significantly reactive nucleophiles in $S_N2$ reactions on the C-atom.

Since the basicity of the investigated nucleophiles is correlated with the nucleophilicity of nucleophiles in the substitution reaction on the S-atom of the S-S bond of TMTD, cyanide and amide ions used in these reactions gave higher product yields, while the iodide ion as a weak base did not provide a significant yield ($CN^-$—96.8%, $NH_2^-$—50.4%, $I^-$—25.0%). The reaction probably occurred by the iodide ion due to its polarizability, which is responsible for the good overlap of its polarizable 5p-orbital with the 3p-orbital of sulfur in TMTD. One should certainly take into account the fact that in the examined reactions, the

cyanide ion reacted in a weakly acidic medium, the iodide ion in an alcoholic medium, and the amide ion in an inert organic solvent (xylene). The highest yields were obtained under these reaction conditions. The influence of protic solvents, where they can be applied in this synthesis reaction, played an essential role in the initial stage of the reaction, which helped the heterolysis of the S-S bond in TMTD by hydrogen interaction.

In the second step of the reaction, nucleophilic substitution occurrs by a nucleophilic attack on the carbon atom of the dimethyl thiocarbamyl intermediate, which was formed in the first step of the reaction, with the dimethyl dithiocarbamate anion. Considering that the dimethyl dithiocarbamate anion is a strong nucleophile, the reaction occurs if, in the first step of the reaction, there is a thiophilic heterolysis of the S-S bond in TMTD and the formation of these intermediates.

### 3.1.4. Results of the Synthesis of the Sodium Salt of Dimethyldithiocarbamic Acid

As part of the study of the mechanism of nucleophilic heterolysis of the S-S bond in TMTD, the sodium salt of dimethyldithiocarbamic acid was synthesized to obtain TMTS. This salt was isolated as an intermediate in the first stage of the reaction so that the structural analysis results of the isolated intermediate could be compared with the results of the synthesized compound. The yield of the obtained sodium salt of dimethyldithiocarbamic acid was 46.12 g (92.4%), and the melting temperature was 121–122 °C (literature data 120–122 °C [22,23]).

### 3.2. Overview of the Experimental Results That Confirmed the Reaction Mechanism of TMTS Synthesis

The reaction of sulfur elimination from the S-S bond in TMTD by cyanide, amide, and iodide ions takes place in two stages, whereby the product TMTS is obtained. In the first step of the reaction, dimethyl dithiocarbamate anion and the corresponding dimethyl thiocarbamyl sulfencyanide were formed (reaction with cyanide). In the second step, nucleophilic substitution occurred on the C-atom of sulfencyanide using the resulting dimethyl dithiocarbamate anion, resulting in the separation of the TMTS product and the thiocyanate ion. Considering that the intermediate dimethyl dithiocarbamate anion was formed in the first step of the reaction, the method of "capturing" the formed intermediate using the zinc $Zn^{2+}$ ion was applied [22], whereby the zinc salt of dimethyldithiocarbamic acid was obtained in the form of a suspension (Scheme 6). The suspension was filtered, and the filtrate was dried and analyzed to confirm the structure of the separated zinc bis-dimethyl dithiocarbamate compound.

The second stage of the synthesis reaction was proven by the identification of the TMTS reaction product and the isolated thiocyanate by-product. 12.81 g (0.042 mol) of zinc dimethyl dithiocarbamate was obtained, and its structure was confirmed by IR and MS analysis. The melting temperature of the separated compound was 249–256 °C (literature data: 248–257 °C [23]).

Isolation of the dimethyl dithiocarbamate ion in the experiments using the amide ion was also proved by confirming the structure of the obtained zinc dimethyl dithiocarbamate by IR spectroscopy, elemental analysis, and the atomic absorption spectroscopy (AAS) method. Using the amide ion as a thiophile, 5.34 g (0.017 mol) of zinc dimethyl dithiocarbamate with a melting temperature of 248–257 °C was obtained. In an experiment in which an iodide ion was used as a thiophile, 3.05 g (0.009 mol) of zinc dimethyl dithiocarbamate with a melting temperature of 246–257 °C was obtained. The IR spectra of zinc dimethyl dithiocarbamate isolated in the previous two experiments are shown in Figure S4.

The examined reaction of TMTS synthesis takes place as follows (Scheme 7): The reaction of dimethylamine (1) and carbon disulfide (2) produced the amine salt of dimethyldithiocarbamic acid (3), from which TMTD (4) was obtained by oxidation with hydrogen peroxide. This reaction takes place successively so that the resulting amine salt undergoes oxidation to TMTD with the separation of amine (1), which reacts with the present carbon-disulfide (2) in the reaction mixture. Specifically, at the beginning of the reaction, in the step of

adding carbon disulfide, half of the added amount reacted with the amine and the other remained in the reaction flask. This is due to the amine consumption required to obtain the dithiocarbamate amine salt. When hydrogen peroxide was added, the amine salt was oxidized to TMTD with the amine release, which immediately reacted with the carbon disulfide present to regenerate the amine salt of dimethyldithiocarbamic acid. The addition of hydrogen peroxide was then continued until complete oxidation of the amine salt in TMTD. In the second step of the reaction, sulfur was eliminated from the disulfide bond of TMTD with the help of the cyanide ion, whereby TMTS was formed by the separation of potassium thiocyanate. The reaction occurred when the nucleophile performed heterolysis on the disulfide bond in TMTD, with the release of dimethyl thiocarbamoyl thiocyanate (5) and dimethyl dithiocarbamate anion (6). Dimethyl dithiocarbamate anion (6), as a nucleophile, reacted with dimethyl thiocarbamoyl thiocyanate (5), whereby TMTS (7) was formed with the isolation of thiocyanate (8). The determination of the thiocyanate ion, following the procedure described in the Supplementary Material, undoubtedly confirmed the suggested mechanism (it stoichiometrically relates to produced TMTS).

**Scheme 6.** Synthesis of TMTS from TMTD using $CN^-$, $NH_2^-$, and $I^-$ nucleophiles and isolation of zinc dimethyl dithiocarbamate as an intermediary.

The kinetics of the reaction were studied to design the industrial process. The reaction takes place by adding potassium cyanide to a solution of TMTD in methanol. In order to study the kinetics of the reaction, TMTD concentrations in methanol were changed from $4.46 \times 10^{-5}$ to $10.00 \times 10^{-5}$ mol/dm$^3$, and potassium cyanide from $1.30 \times 10^{-4}$ to $15.30 \times 10^{-4}$ mol/dm$^3$. The temperature at which the reaction takes place changes in the range of 5.0–25.0 °C. The reaction of TMTD with cyanide ions takes place in two stages. The reaction is in the first order with respect to TMTD and is more and close to the first order with respect to the $CN^-$ ion, and this conclusion is in accordance with literature data [24]. The fact that the reaction takes place in two stages in which the concentration of intermediates increases was determined by spectral analysis and isolation. A sample of the reaction mixture was taken after a certain reaction time and analyzed. UV spectroscopy

gives the spectrum of the mixture of TMTD and TMTS, as well as dimethyl dithiocarbamate. It is clear that the reaction does not take place in one step and that intermediates are present in variable concentrations.

**Scheme 7.** Reaction mechanism of TMTS synthesis.

*3.3. Characterization Results*

3.3.1. Characterization of Synthesized Products and Isolated Intermediates

In all the presented experiments, the synthesized products were identified based on the obtained UV, IR, and MS spectra. All spectra show characteristic bands of TMTS.

Tetramethyl Thiuram Disulfide (TMTD)

White to yellow powder, yield 99.2%, m.p. 152–156 °C; elemental analysis: Calcd. for $C_6H_{12}N_2S_4$ (Mw = 240.42 g mol$^{-1}$): C, 29.98; H, 5.03; N, 11.65%; Found: C, 30.01; H, 4.98; N, 11.41%; UV/Vis (nm): 250 and 230 nm; IR, cm$^{-1}$: 2932 cm$^{-1}$ $\nu$(C–H stretching of –CH$_3$ group), 1506 cm$^{-1}$, $\nu$(amide II), 1402 cm$^{-1}$, 1378 cm$^{-1}$, $\delta$(C–H bending of –CH$_3$ group), 1237 cm$^{-1}$ $\nu$(amide III), 1152 cm$^{-1}$ $\nu$(C=S stretching), 1040 cm$^{-1}$ $\nu$(C-N stretching), 978 cm$^{-1}$, 971 cm$^{-1}$ $\nu$(C-S stretching), 563 cm$^{-1}$, 442 cm$^{-1}$ $\nu$(S-S); $^1$H NMR (500 MHz, DMSO-$d_6$, $\delta$ ppm) = 3.58 (s, 12H, CH$_3$); $^{13}$C NMR (125 MHz, DMSO-$d_6$, $\delta$ ppm) $\delta$ = 44.7 (4CH$_3$), 195.1 (2C=S). The NMR results are in accordance with the literature data [6,25]; MS: calculated for $C_6H_{12}N_2S_4$, (m/z): 239.99, observed: 240.

Tetramethyl Thiuram Monosulfide (TMTS)

Yellow powder, yield 96.57%, m.p. 105–109 °C; elemental analysis: Calcd. for $C_6H_{12}N_2S_3$ (Mw = 208.02 g mol$^{-1}$): C, 34.59; H, 5.81; N, 13.45%; Found: C, 35.02; H, 5.77; N, 13.20%; UV/Vis (nm): 275 and 240 nm; IR, cm$^{-1}$: 2930 cm$^{-1}$ $\nu$(C–H stretching of –CH$_3$ group), 1502, 1520 cm$^{-1}$, $\nu$(amide II), 1402 cm$^{-1}$, 1412, 1420, 1380 cm$^{-1}$, $\delta$(C–H bending of –CH$_3$ group), 1234 cm$^{-1}$, 1250 $\nu$(amide III), 1151 cm$^{-1}$ $\nu$(C=S stretching), 1053 cm$^{-1}$ $\nu$(C-N stretching), 998 cm$^{-1}$ and 963 cm$^{-1}$ $\nu$(C-S stretching), 871–863 cm$^{-1}$, 551 cm$^{-1}$–440 cm$^{-1}$ $\nu$(S-C-S); $^1$H NMR (500 MHz, DMSO-$d_6$, $\delta$ ppm) = 3.59 (s, 12H, CH$_3$); $^{13}$C NMR (125 MHz, DMSO-$d_6$, $\delta$ ppm) $\delta$ = 44.9 (4CH$_3$), 186.2 (2C=S). The NMR results are in accordance with the literature data [6,25]; MS: calculated for $C_6H_{12}N_2S_3$, (m/z): 208.02, observed: 208.

Sodium Salt of Dimethyl Dithiocarbamic Acid

Yellow solid, m.p. 121–122 °C (literature data: 120–122 °C [23]); elemental analysis: calcd. for $C_3H_6NNaS_2$ (Mw = 143.20 g mol$^{-1}$): C, 25.16; H, 4.22; N, 9.78%; found: C, 24.99; H, 4.25; N, 10.01%; IR, cm$^{-1}$: 2852 cm$^{-1}$ $\nu$(C–H stretching of –CH$_3$ group), 1480 cm$^{-1}$, 1488 cm$^{-1}$, $\delta$(C–H bending of –CH$_3$ group), 1350 cm$^{-1}$, 1237 cm$^{-1}$ $\nu$(amide III), 1100 cm$^{-1}$ $\nu$(C=S stretching), 1020 cm$^{-1}$, 981 cm$^{-1}$ $\nu$(C-S stretching), 543 cm$^{-1}$, 432 cm$^{-1}$ $\nu$(S-S; $^1$H NMR (500 MHz, DMSO-$d_6$, $\delta$ ppm) = 3.5 (s, 12H, CH$_3$); $^{13}$C NMR (125 MHz, DMSO-$d_6$, $\delta$ ppm) $\delta$ = 44.3 (4CH$_3$), 199.9 (C=S). The NMR results are in accordance with the literature data [26]. MS: calculated for $C_3H_6NNaS_2$, (m/z): 142.98, observed: 121.

Zinc Dimethyl Dithiocarbamate (Ziram)

White solid powder, m.p. 249–256 °C (literature data: 242–255 °C [27], 248–257 °C [23]) elemental analysis: calcd. for $C_{12}H_{24}N_4S_8Zn_2^{6-}$ (Mw = 611.60 g mol$^{-1}$): C, 23.57; H, 3.96; N, 41.94; Zn, 21.38%; Found: C, 23.98; H, 4.12; N, 42.05; Zn, 21.29%; IR, cm$^{-1}$: 2930 cm$^{-1}$ ν(C–H stretching of –CH$_3$ group), 1512 cm$^{-1}$, ν(amide II), 1436 cm$^{-1}$, 1389 cm$^{-1}$, δ(C–H bending of the –CH$_3$ group), 1241 cm$^{-1}$ ν(amide I), 1143 cm$^{-1}$ ν(C=S stretching), 1049 cm$^{-1}$, 971 cm$^{-1}$ ν(C-S stretching), 564 cm$^{-1}$ ν(S-S); $^1$H NMR (500 MHz, DMSO-$d_6$, δ ppm) = 3.38 (s, 12H, CH$_3$); $^{13}$C NMR (125 MHz, DMSO-$d_6$, δ ppm) δ = 44.8 (4CH$_3$), 199.8 (2C=S). The NMR results are in accordance with the literature data [28].

*3.4. Industrial Process of TMTS Production*

The industrial procedure for the production of TMTS was carried out using the process equipment shown in Figure 1. In a reactor with a volume of 5 m$^3$ (position 1), 1.5 m$^3$ of an azeotropic mixture of isopropyl alcohol-water (87.7–12.3%) was introduced from a dispenser (position 3), and 0.437 m$^3$ (4.16 kmol) of 50.0% dimethyl amine solution from the dispenser (position 5). After starting the stirrer, 0.256 m$^3$ (4.16 mol) of 98.0% carbon disulfide was added from doser CS$_2$ (position 4) over 0.5 h while maintaining the temperature in the range of 28–35 °C, provided by circulating cooling water. At the end of the reaction, the pH of the reaction mixture was 6.5. At this point, 0.536 m$^3$ of a 13.2% hydrogen peroxide solution (prepared by diluting 0.178 m$^3$ (2.08 mol) of 35.0% hydrogen peroxide with 0.406 m$^3$ of an isopropyl alcohol/water azeotropic mixture) from the H$_2$O$_2$ dispenser (position 6) were added to the reaction for 0.5 h at 35–40 °C. After the completion of the reaction, the reaction mixture turned yellowish due to the suspended TMTD particles. At the end of the first stage of the synthesis reaction, 231.11 kg (4.16 mol) of ammonium chloride and a 20% aqueous solution of potassium cyanide (285.42 kg of potassium cyanide) were added to the suspension of the obtained TMTD (21%) dissolved in 1.068 m$^3$ of water (4.16 mol) from the KCN solution dispenser (position 8). The entire amount of potassium cyanide was added for 2 h. Then, the reaction mixture was stirred for another hour while maintaining the reaction temperature of 50 °C. After completion of the reaction, the reaction mixture was filtered on a vacuum filter device (position 10). The filtrate was collected in tanks (positions 11 and 12) and used for the following TMTS synthesis reaction. The filter cake was washed with water (negative SCN$^-$ test), and the crystals were dried in a vacuum dryer (position 13) at 60 °C until the moisture content was below 0.5%. The dried crystals were ground in the mill section (position 14) to the required granulation and forwarded to formulation and packaging. After the filtrate was used as a reaction medium for the subsequent synthesis, it was transferred to rectification (position 15) and stored in a tank (position 9). The results of the semi-industrial synthesis of TMTS as a function of the amounts of reactants are given in Table 6. Figure 1 presents the technological scheme of the industrial process of TMTS production.

It is generally accepted that two valuable measures of chemical processes' (potential) environmental acceptability are the E factor, defined as the mass ratio of waste to the desired product, and the atom efficiency. The atom efficiency or atom economy concept is an extremely useful tool for rapid evaluation of the amounts of waste that will be generated by alternative processes. It is calculated by dividing the product's molecular weight by the sum total of the molecular weights of all substances formed in the stoichiometric equation for the reaction involved. For bulk chemicals manufactured in hundreds of thousands to millions of tons per year, tolerable E factors typically range from 1 to 5 [29]. In the fine chemical and specialty chemical industries, where annual quantities are usually measured as a few thousand tons per year, E factors up to around 500 may be acceptable if the product's value is high enough to justify the cost of treating and disposing of waste [30]. The E-factor of the industrial process presented in this manuscript is 0.47. The leftover reactant that can be easily reclaimed and recycled in the process is not included as waste, whereas the reactant that cannot be salvaged is counted in the waste.

**Table 6.** The results of the semi-industrial synthesis of TMTS.

| Batch | Reactants [a] | | | | | Reaction Conditions | | Product | | |
|---|---|---|---|---|---|---|---|---|---|---|
| | $(CH_3)_2NH$ $m^3$ (kmol) | $CS_2$ $m^3$ (kmol) | $H_2O_2$ $m^3$ (kmol) | KCN $m^3$ (kmol) | $NH_4Cl$ kg (kmol) | Time [b] (h) | T [c] (°C) | Yield (%) | m.p. (°C) | Purity (%) |
| 1 | 0.437 (4.16) | 0.256 (4.16) | 0.536 (2.08) | 1.353 (4.16) | 231.1 (4.16) | 1.0/2.0 | 28–35 35–40 50 | 94.0 | 105–108 | 99.2 |
| 2 [d] | 0.437 (4.16) | 0.268 (4.36) | 0.562 (2.18) | 1.353 (4.16) | 231.1 (4.16) | 1.5/2.5 | 20–30 30–35 50 | 95.0 | 105–108 | 99.3 |
| 3 | 0.437 (4.16) | 0.437 (4.16) | 0.536 (2.08) | 1.353 (4.16) | 242.1 (4.36) | 1.0/2.5 | 25–30 35–40 50 | 94.6 | 105–109 | 99.2 |

[a] Solvent for synthesis: azeotropic mixture of iPrOH/$H_2O$ (87.7/12.3%), 1.5 $m^3$. [b] I step of the reaction: TMTD synthesis; II step: TMTS production. [c] I step of the reaction: $CS_2$ dosage; II step: $H_2O_2$ dosage; III step: KCN dosage. [d] The filtrate from batch I was used in batch II as a solvent, iPrOH/$H_2O$ (70.4/29.6%).

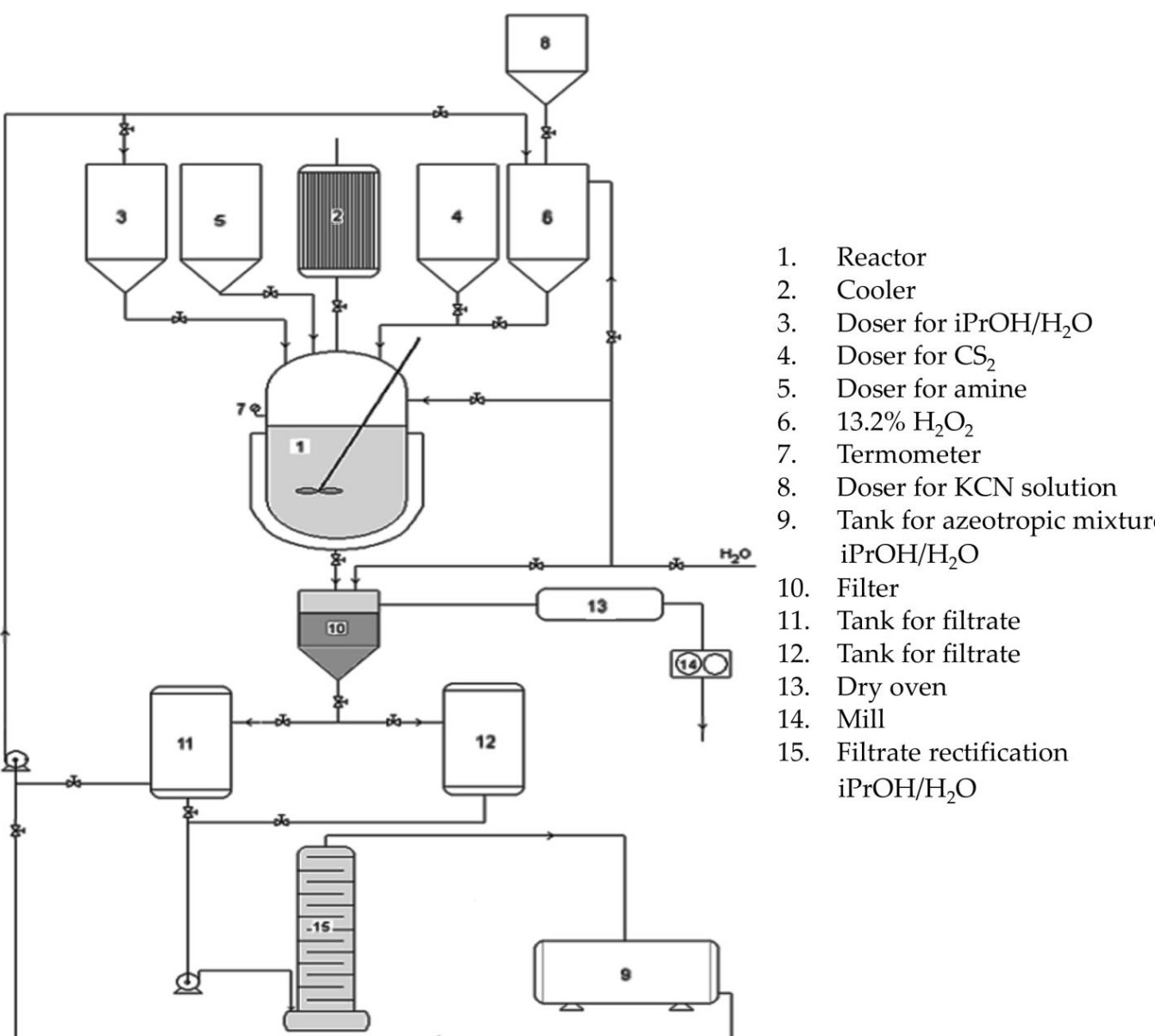

1. Reactor
2. Cooler
3. Doser for iPrOH/$H_2O$
4. Doser for $CS_2$
5. Doser for amine
6. 13.2% $H_2O_2$
7. Termometer
8. Doser for KCN solution
9. Tank for azeotropic mixture iPrOH/$H_2O$
10. Filter
11. Tank for filtrate
12. Tank for filtrate
13. Dry oven
14. Mill
15. Filtrate rectification iPrOH/$H_2O$

**Figure 1.** The technological scheme for TMTS industrial production.

Safety measures were achieved by installing three scrubbers: one with 100 dm$^3$ of recirculating 10% lead (II) acetate for $H_2S$ removal; the second with 100 dm$^3$ of recirculating 2 mol/dm$^3$ KOH water solution mainly for HCN and $CS_2$ removal; and the third with 2 mol/dm$^3$ KOH ethanol solution for $CS_2$ removal in the form of potassium ethyl xanthate.

Results of the characterization of TMTD and TMTS (industrial level) are given in Table 7.

**Table 7.** The physico-mechanical characteristics of the accelerators TMTS and TMTD.

| Accelerator | TMTS | TMTD |
|---|---|---|
| Melting point, °C | 105–109 | 152–156 |
| Appearance | Yellow powder | White powder |
| Bulk material in a loose state (g/cm$^3$) | 0.220 | 0.260 |
| Moisture at 105 °C (%) | 0.029 | 0.03 |
| Ash at 600 °C (%) | 0.022 | 0.020 |
| Ash content insoluble in HCl | 0.014 | 0.012 |
| The mass loss at 85 °C, % | 0.024 | 0.021 |
| pH of 5% suspension | 6.8 | 6.9 |
| Sieve analysis (no residual mater.) | 150 μm | 150 μm |
| Metal content, ppm | | |
| Cu | - | - |
| Mn | - | - |
| Fe | 3.44 | 3.52 |

### 3.5. Results of the LA/EG/PET/PG/LA Plasticizer Synthesis

Results of the characterization of 2-((4-oxopentanoyl)oxy)ethyl (2-((4-oxopentanoyl)oxy) propyl) terephthalate—LA/PG/PET/EG/LA are given in the Supplementary Material.

### 3.6. Rubber Products Properties

Processability is defined as the behavior and interactions of polymer, filler, oil, and other additives during the various processing stages, such as mixing, extrusion, calendaring, and molding processes. It depends on the cure characteristics and rheological properties of the rubber compound, as well as other components of filler, such as oil and additives". The main processability parameters are cure time, scorch time, viscosity, and extrusion rate.

In order to investigate the influence of the synthesized plasticizer and accelerator on the physico-mechanical and mechanical properties of NBR and SBR rubber mixtures, the replacement of DOTP with synthesized waste/biobased PET plasticizers, i.e., LA/EG/PET/PG/LA, was performed. Vulcanization is a cross-linking process in which individual rubber molecules (polymers) are converted into a three-dimensional network of interconnected polymer chains through chemical cross-links of sulfur.

The period of time before vulcanization starts is referred to as scorch time (ts$_2$), and it is an important characteristic of the vulcanization process. It is evident from the obtained results that there is a noticeable change concerning most of the cure properties obtained from the rheographs, such as scorch time, optimum cure time (tc$_{90}$), maximum rheometric torque (M$_H$), and minimum torque (M$_L$). Premature vulcanization during processing is considered scorch. The majority of accelerator chemistry took place in the scorch delay period or the induction region. Generally, scorch time (ts$_2$) is defined as the time to reach a two-unit increase in torque above the minimum. An accelerator is a chemical added to a rubber compound to increase the speed of vulcanization and permit vulcanization to proceed at a lower temperature and with greater efficiency. The accelerator also decreases the quantity of sulfur necessary for vulcanization and thus improves the 'aged' properties of the rubber vulcanizates. In general, thiuram accelerators TMTS and TMTD during vulcanization offer short scorch times, as TMTD is designated as an ultra-fast accelerator.

Mooney viscosity measures the stiffness of the uncured compounds, otherwise known as the compound's viscosity. The Mooney viscometer consists of a rotor rotating at 2 rpm in a closed, heated cavity chamber filled with uncured rubber. A shearing action develops between the compound and the rotor, and the resulting torque (the resistance of the rubber to the turning rotor) is measured in arbitrary units called Mooney units, which is directly related to torque. The larger the number, the higher the viscosity, e.g., ML 1 + 4 (100 °C), where M indicates Mooney, L indicates that a large rotor was used (S for a small rotor), 1 indicates one minute of preheating time, 4 refers to the time in minutes after starting the rotor, and 100 °C is the test temperature (Tables 8 and 9).

**Table 8.** The rheological properties of NBR after vulcanization at 150 °C for 15 min.

| Rubber Blend | $NBR_0$ | $NBR_1$ | $NBR_2$ | $NBR_3$ |
|---|---|---|---|---|
| Accelerator | TMTD | TMTD | TMTS | TMTS |
| Plasticizer | DOTP | LA/PG/PET/EG/LA | DOTP | LA/PG/PET/EG/LA |
| $M_H$ (dNm) | 52 | 49 | 51 | 49 |
| $M_L$ (dNm) | 9.5 | 9.2 | 8.7 | 8.5 |
| $t_{S2}$ (min) | 2.86 | 2.82 | 3.18 | 3.11 |
| $t_{C90}$ (min) | 5.78 | 5.82 | 6.08 | 6.96 |
| Mooney ML (1 + 4), 100 °C | 53 | 62 | 60 | 58 |

**Table 9.** The rheological properties of SBR after vulcanization at 170 °C for 15 min.

| Rubber Blend | $SBR_0$ | $SBR_1$ | $SBR_2$ | $SBR_3$ |
|---|---|---|---|---|
| Accelerator | TMTD | TMTD | TMTS | TMTS |
| Plasticizer | DINP | LA/PG/PET/EG/LA | DINP | LA/PG/PET/EG/LA |
| $M_H$ (dNm) | 42 | 39 | 43 | 38 |
| $M_L$ (dNm) | 7.5 | 7.7 | 6.2 | 6.8 |
| $t_{S2}$ (min) | 3.36 | 3.11 | 4.42 | 4.28 |
| $t_{C90}$ (min) | 7.17 | 7.33 | 7.48 | 7.51 |
| Mooney ML (1 + 4), 100 °C | 51 | 59 | 54 | 53 |

3.6.1. Physico-Mechanical Properties of Rubber Blends

After standing for 24 h, the rubber samples were cut into a defined specimen and subjected to study according to standard tests (Table S3). The obtained results of the physico-mechanical properties of NBR and SBR-based mixtures after vulcanization are given in Tables 8–13. The results of the rheological properties after vulcanization are presented in Tables 8 and 9.

The scorch time was considered to be the time before 2% vulcanization occurred. This is important, and it is generally the scorch safety time. A small scorch safety value for a vulcanizate causes premature vulcanization due to uneven cross-links in the thick vulcanizate. Hence, better scorch safety and shorter curing times are needed for thick vulcanizates. Generally, thiuram compounds have lower scorch safety but higher curing rates. A faster curing rate is economical for the rubber industry. However, a shorter curing time with a longer scorch safety time is most important for technological vulcanizates with excellent physical properties.

**Table 10.** The mechanical properties of NBR after vulcanization at 150 °C for 15 min.

| Rubber Blend | NBR$_0$ | NBR$_1$ | NBR$_2$ | NBR$_3$ |
|---|---|---|---|---|
| Accelerator | TMTD | TMTD | TMTS | TMTS |
| Plasticizer | DOTP | LA/PG/PET/EG/LA | DOTP | LA/PG/PET/EG/LA |
| Shore hardness (°ShA) | 63 | 65 | 66 | 64 |
| Tensile strength (MPa) | 17.88 | 18.21 | 15.08 | 17.24 |
| Stress at 100% elongation (MPa) | 1.62 | 1.82 | 1.68 | 1.83 |
| Stress at 300% elongation (MPa) | 11.82 | 12.22 | 10.38 | 11.23 |
| Elongation at break (%) | 410 | 420 | 375 | 396 |
| Resilience (%) | 42 | 41 | 38 | 37 |
| Density (g/cm$^3$) | 1.25 | 1.24 | 1.26 | 1.23 |

**Table 11.** The mechanical properties of SBR after vulcanization at 170 °C for 15 min.

| Rubber Blend | SBR$_0$ | SBR$_1$ | SBR$_2$ | SBR$_3$ |
|---|---|---|---|---|
| Accelerator | TMTD | TMTD | TMTS | TMTS |
| Plasticizer | DINP | LA/PG/PET/EG/LA | DINP | LA/PG/PET/EG/LA |
| Shore hardness (°ShA) | 55 | 58 | 49 | 54 |
| Tensile strength (MPa) | 11.43 | 13.08 | 8.6 | 9.7 |
| Stress at 100% elongation (MPa) | 1.21 | 1.78 | 1.33 | 1.22 |
| Stress at 300% elongation (MPa) | 3.59 | 4.78 | 2.33 | 2.81 |
| Elongation at break (%) | 726 | 754 | 738 | 772 |
| Resilience (%) | 43 | 44 | 40 | 41 |
| Compression set, 70 h at 100 °C (%) | 25.2 | 34.1 | 16.5 | 18.8 |
| Density (g/cm$^3$) | 1.11 | 1.12 | 1.11 | 1.10 |

**Table 12.** The change of NBR-based product properties after aging in the air at 100 °C for 70 h.

| Rubber Blend | NBR$_0$ | NBR$_1$ | NBR$_2$ | NBR$_3$ |
|---|---|---|---|---|
| Accelerator | TMTD | TMTD | TMTS | TMTS |
| Plasticizer | DINP | LA/PG/PET/EG/LA | DINP | LA/PG/PET/EG/LA |
| Shore hardness (°ShA) | +6 | +3 | +5 | +2 |
| Tensile strength (MPa) | −1.88 | −1.21 | −1.54 | −1.08 |
| Elongation at break (%) | −20.7 | −13.4 | −14.5 | −12.6 |
| Swelling for 72 h at 70 °C in oil 15 W–40 | | | | |
| Shore hardness (°ShA) | −3 | −2 | −4 | −2 |
| Swelling, % | 1.87 | 1.11 | 2.21 | 1.22 |

**Table 13.** The change of vulcanizate properties after aging at 100 °C for 24 h.

| Rubber Blend | $SBR_0$ | $SBR_1$ | $SBR_2$ | $SBR_3$ |
|---|---|---|---|---|
| Accelerator | TMTD | TMTD | TMTS | TMTS |
| Plasticizer | DINP | LA/PG/PET/EG/LA | DINP | LA/PG/PET/EG/LA |
| Shore hardness (°ShA) | +3 | +2 | +3 | +1 |
| Tensile strength (MPa) | −1.48 | −1.25 | −1.45 | −0.98 |
| Elongation at break (%) | −18.7 | −10.6 | −14.8 | −8.6 |

3.6.2. Aging

The oxidative aging of the rubber vulcanizates is of utmost importance for practical purposes. In the aging experiment, vulcanizates obtained at optimum cure are aged at $70 \pm 1$ °C in an oven provided with forced air circulation for 1–5 days. Each day, the specimens thus aged are kept for another 24 h at room temperature before the modulus and tensile strength are measured. Changes in vulcanizate properties after aging are presented in Tables 12 and 13.

**4. Discussion**

TMTS was prepared by simple and efficient one-pot synthesis from dimethylamine, carbon disulfide, potassium cyanide, and ammonium chloride as catalysts in recycled isopropanol/water azeotrope as solvent. Methoxide, ethoxide, iodide, and the amide ion were also used as nucleophiles to study and confirm the mechanism of the reaction. The results obtained by the intermediates' isolation procedures and quantifying the generated thiocyanate ions proved the two-step reaction mechanism: oxidation of the amine salt of the dimethyldithiocarbamic acid to TMTD by hydrogen peroxide and sulfur elimination from the TMTD disulfide bond. Potassium cyanide appeared to be the most efficient nucleophile. Dimethyldithiocarbamate anion and dimethyl thiocarbamoyl sulfencyanide were formed in the first step of the reaction. Hence the formed intermediate was captured by $Zn^{2+}$, and the zinc salt of dimethyldithiocarbamic acid was obtained as a suspension. In the second step, nucleophilic substitution at the C-atom of sulfencyanide with the dimethyl dithiocarbamate anion was carried out, followed by the separation of the TMTS product and the thiocyanate ion. The second step of the synthesis reaction was proven by the identification of the TMTS reaction product and the isolated thiocyanate by-product, whose concentration in the filtrate was determined. Isolated thiocyanate can also be further used as the reactant for the synthesis of other sulfur-containing compounds [31]. The advantages of the process included the simplicity of operation, mild reaction conditions, solvent recycling, high yields, and applicability to the industrial level. The industrial trial production of TMTS showed significant yields of the synthesized product (94–95%) with a high degree of purity (99.2–99.3%). The synthesis reaction in batch 2 was performed with an excess of reactants $CS_2$ and $H_2O_2$ in the amount of 5% and with an extended reaction time of 2.5 h, which resulted in a slight increase in conversion to 95%. Also, in batch 3, a yield of 94.6% was achieved when the amounts of reactants KCN and $NH_4Cl$ were increased by 5% each and the dosing time of KCN was extended by half an hour. Also, by using the filtrate separated on the vacuum filter from batch 1 in batch 2, it can be noticed that there were no significant deviations in terms of conversion, which meant that the reaction medium was recyclable. After two batches of synthesis, it was necessary to redistill the solvent because of increased water content, so the $iPrOH/H_2O$ ratio of 70.4/29.6% after the second batch was reduced to 50.6/49.4%.

The influence of the synthesized plasticizers and accelerators on the rheological and mechanical properties of NBR and SBR rubber mixtures was investigated by replacing DOTP with synthesized waste/biobased PET plasticizers, i.e., LA/EG/PET/PG/LA. As can be seen from the rheological results, the rubber products with synthesized **LA/PG/PET/EG/LA plasticizer**, based on the chemical recycling of waste PET resources and biobased raw materials

and using the accelerator TMTD, have slightly better physical and mechanical properties: hardness, modulus, and breaking strength increase, while elongation at break decreases. On the other hand, these results clearly show that glycolizates from PET in NBR mixtures act as plasticizers, while after vulcanization, they act as reinforcing materials. Due to the mentioned results of physico-mechanical tests on rubber products, the obtained materials can be compared with standard, commercial plasticizers (DOTP), but PET-based plasticizers are preferred since they act as plasticizers during the processing and as reinforcing materials after vulcanization.

## 5. Conclusions

In this work, styrene-butadiene (SBR) and butadiene-nitrile (NBR) rubber were used as a network precursor to prepare rubber blends with tetramethyl thiuram disulfide (TMTD) and tetramethyl thiuram monosulfide (TMTS) as accelerators combined with plasticizers obtained from PET recycling to produce environmentally friendly materials with improved dynamic-mechanical and mechanical properties. The influence of synthesized accelerators and plasticizers on blend characteristics and the obtained elastomer nanocomposites was examined and compared with that of the commercial plasticizer DOTP. Additionally, a new laboratory procedure for the synthesis of TMTS was defined by performing the one-pot synthesis from TMTD by eliminating sulfur from the persulfide S-S bond using potassium cyanide and ammonium chloride as catalysts, in the isopropanol/water solvent (87.7–12.3%) regenerated by rectification after recycling.

The reaction mechanism of the synthesis of TMTS from TMTD was confirmed by using different nucleophiles (amide, iodide, and cyanide ions), while ethoxide and methoxide ions did not react under the examined conditions. In the first step of the reaction, the intermediate-dimethyl dithiocarbamate anion is substituted by the nucleophile used and isolated by a trapping agent ($Zn^{2+}$ ion), whereby zinc dimethyl dithiocarbamate suspension is formed, and its structure is confirmed by appropriate instrumental methods.

Based on the reproducible results obtained in the experimental part of this manuscript, an industrial trial production of TMTS was carried out. The optimal industrial procedure for the production of TMTS was defined.

The rubber products, with commercial plasticizer DINP and synthesized **LA/PG/PET/ EG/LA**, based on bio- and chemical recycling of waste PET resources and using accelerator TMTD, have better physical and mechanical properties: hardness, modulus, and breaking strength increase, while elongation at break decreases. These results indicated that PET glycolizates in NBR mixtures act as plasticizers and, after vulcanization, as reinforcing materials.

**Supplementary Materials:** The following supporting information can be downloaded at: https: //www.mdpi.com/article/10.3390/pr11041033/s1. Figure S1: FTIR spectrum of PG and LA; Figure S2: The most convenient structure of synthesized plasticizer LA/PG/PET/EG/LA; Figure S3: Dependence of absorbance on TMTS concentration; Figure S4: Zinc-bis(dimethyl dithiocarbmate) structure; Figure S5: FTIR spectrum of Ziram; Figure S6: FTIR spectra of the synthesized waste PET-based plasticizers (LA/PG/PET/EG/LA); Figure S7: (a) [1]H NMR and (b) [13]C NMR spectra of 2-((4-oxopentanoyl)oxy)ethyl 2-((4-oxopentanoyl)oxy)propyl terephthalate (LA/EG/PET/PG/LA); Table S1: Rubber products composition with 20 phr of plasticizers based on NBR 28% acrylonitrile; Table S2: Rubber products composition with 21 phr of plasticizers based on SBR rubber; Table S3: The standard methods used for the determination of a physico-mechanical characteristic of a rubber product; Table S4: HV and AV values and results of elemental analysis of synthesized intermediate and LA/PG/PET/EG/LA plasticizers. Refs. [32,33] are cited in Supplementary Materials.

**Author Contributions:** Conceptualization, M.M. (Milutin Milosavljević) and A.M.; methodology, A.M. and M.R.; formal analysis, S.M. (Smiljana Marković); investigation, G.M.; data curation, S.M. (Svetomir Milojević) and S.M. (Smiljana Marković); writing—original draft preparation, G.M. and S.M. (Svetomir Milojević); writing—review and editing, M.M. (Milena Milošević) and M.R.; supervision, M.M. (Milutin Milosavljević); All authors have read and agreed to the published version of the manuscript.

**Funding:** This research was funded by the Ministry of Education, Science, and Technological Development of Serbia (Project Number 451-03-47/2023-01/200135, Project Number 451-03-47/2023-01/200169, Project Number 451-03-47/2023-01/200026, and Project Number 43007).

**Data Availability Statement:** Not applicable.

**Conflicts of Interest:** The authors declare no conflict of interest. The funders had no role in the design of the study, in the collection, analysis, or interpretation of data, in the writing of the manuscript, or in the decision to publish the results.

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
