# Peer review of "One-Pot Syntheses of PET-Based Plasticizer and Tetramethyl Thiuram Monosulfide (TMTS) as Vulcanization Accelerator for Rubber Production"

_processes, doi:10.3390/pr11041033_

Round 1

Reviewer 1 Report

The paper by Milentijević and co-workers deals with the tetramethyl thiuram monosulfide synthesis (TMTS) to apply as accelerator and plasticizer towards the preparation of rubber blends with styrene-butadiene (SBR) and butadiene-nitrile (NBR) rubber. The reaction yields of TMTS were up to 97%, including the one-pot approach. The work brings a catalytic methodological study together with scaling-up. However, it needs some clarification and discussion before been published.

 - Authors mentioned the importance of sustainable routes for the reactions. However, the E-factor should be calculated and included in the work for better analyses of scale up.

- Authors defined that the basic nucleophiles are more decisive for the reactions but did not explained the case of amide ion. Why does it is less reactive nucleophile than the CN- ion in the substitution reaction at the S-atom?

- Since authors studied the reaction mechanism, what about the kinetics of the reaction? This factor is also necessary for the scaling up.

- The use of cyanide ions in slightly acid medium can generate HCN, even in small quantities. Authors should quantify and stablish the maximum limit according to the safety rules.

- Specifically in the case of iodide ion as nucleophile, the ammonium iodide was fresh?

- It is necessary to review English grammar.

Author Response

Response to Reviewer 1 Comments

Thank you very much for your letter concerning our paper (processes-2246146) entitled: "One-pot synthesis of PET-based plasticizer and tetramethyl thiuram monosulfide (TMTS) as vulcanization accelerator for rubber production". The title was changed according to Reviewer 2 comment to be more appropriate for the results presented in manuscript.

We appreciate the opportunity to revise our work to be considered for publication in Processes. We have carefully taken the comment of the reviewers into consideration in preparing our revision. Our detailed reply to Reviewers' comments is given below and corrections were marked in yellow. Moreover, all the corrections of grammar and results obtained after checking were done in revised manuscript.

Reviewer 1

The paper by Milentijević and co-workers deals with the tetramethyl thiuram monosulfide synthesis (TMTS) to apply as accelerator and plasticizer towards the preparation of rubber blends with styrene-butadiene (SBR) and butadiene-nitrile (NBR) rubber. The reaction yields of TMTS were up to 97%, including the one-pot approach. The work brings a catalytic methodological study together with scaling-up. However, it needs some clarification and discussion before been published.

Point 1: Authors mentioned the importance of sustainable routes for the reactions. However, the E-factor should be calculated and included in the work for better analyses of scale up.

Response 1: Explanation, definition, and calculated E-factor were introduced in the main manuscript. The E-factor calculated as the mass ratio of waste to desired product is 0.47. For bulk chemicals, tolerable E factors typically range from 1 to 5.  

Point 2: Authors defined that the basic nucleophiles are more decisive for the reactions but did not explained the case of amide ion. Why does it is less reactive nucleophile than the CN- ion in the substitution reaction at the S-atom?

Response 2: Both cyanide ion (CN) and amide ion (NH2) are powerful necleophiles, though amide ion is a much stronger base. The basicity of nitrogen is highest for sp3-hybridized nitrogen (e.g. CH3NH2) and is lowest for sp-hybridized nitrogen (e.g. CH3CN). It is assumed that more basic nucleophile (amide ion) would favor elimination versus substitution. On the other side, strong amide nucleophile is less selective (the IR spectra indicated also side reactions of creation of thiourea in the case of using amide ion as nucleophile).

Moreover, it was stated in the text: "One should certainly take into account that in the examined reactions, the cyanide ion reacts in a weakly acidic medium, the iodide ion in an alkaline medium, and the amide ion in an inert organic solvent (xylene). The highest yields were obtained in these reaction conditions. The influence of protic solvents, where they can be applied in this synthesis reaction, plays an appropriate role in the initial stage of the reaction, in that it helps the hetrolysis of the S-S bond in TMTD by hydrogen interaction ".

The most likely nucleophilic attack mechanism by a nucleophile (cyanide or amide ion) supposes that the reaction occurs in two steps. The first step involves a nucleophilic attack on the sulfur atom from the disulfide bond, whereby in the first case, (CH3)2C(S)S-NH2 sulpheamide is formed, i.e. 1-(aminothio)-N,N-dimethyl-1-thioxomethanamine. In the case of nucleophilic attack of cyanide, (CH3)2C(S)S-CN is formed, i.e. cyanic dimethylcarbamothioic thioanhydride, which contains better and more stable departing group SCN- vs. sulfenamide. In the second step of the reaction, nucleophilic substitution occurs on the C-atom of the dimethylthiocarbamyl intermediate,  with the dimethyldithiocarbamate anion. Because in the case of cyanide attack, intermediate contains better departing group, the better yield was obtained compared to amide anion attack. (Scheme 7 in the manuscript and lines 503-508). Additionally, in this case, thiourea was isolated by eliminating sulfur from dimethyl thiosulfenamide and the decomposition of the present TMTD. These undesirable reactions lead to a lower conversion degree to product (lines 412-414). Details on proposed reaction machanism is given in Robert Earl Davis and Arnold Cohen, J. Am. Chem. Soc, 86 (1964) 440 – 443.

Point 3:  Since authors studied the reaction mechanism, what about the kinetics of the reaction? This factor is also necessary for the scaling up.

Response 3: The kinetics of the reaction was already investigated and published (Robert Earl Davis and Arnold Cohen, J. Am. Chem. Soc, 86 (1964) 440 – 443), but we also studied the kinetics. Our experimental results obtained in the kinetics study were briefly described in the manuscript (lines 514-523).

Point 4:  The use of cyanide ions in slightly acid medium can generate HCN, even in small quantities. Authors should quantify and stablish the maximum limit according to the safety rules.

Response 4: The maximum limit according to the safety rules: Exposure Limits. NIOSH REL. ST 4.7 ppm (5 mg/m3) [skin]. OSHA PEL. TWA 10 ppm (11 mg/m3) [skin] Appendix G (8 hours). Repeated laboratory procedure performed in a closed laboratory reactor using an outlet connected to a glass laboratory gas washer with 5% potassium hydroxide solution (50 ml). Measurement, using a cyanide selective electrode, gave an average 0.075 ppm/30 m3/4 h is far from the limit suggested by appropriate regulative. Moreover, during synthesis, the equilibrium is shifted to cyanide anion formation, which is involved in reaction as nucleophile and removed from the mixture. Also, synthesis in industrial conditions was performed using a properly sealed reactor and also three scrubers with 100 dm3 of recirculating 10% lead(II) acetate for H2S removal, 100 dm3 of recirculating 2 mol dm3 KOH water solution for HCN and CS2 removal (in the form of KCN and tritiocarbonate, respectively), and 2 mol dm3 KOH ethanol solution for CS2 (in the form of potassium ethyl xanthate).

Point 5: Specifically in the case of iodide ion as nucleophile, the ammonium iodide was fresh?

Response 5: The ammonium iodide, used in the reaction, was fresh.

Point 6: It is necessary to review English grammar.

Response 6: English grammar and spelling were done in the revised manuscript

Reviewer 2 Report

This manuscript comprises three topic, as mentioned in Comment 2. Although the part about the synthesis of TMTS and its mechanism is fully studied, others are relatively concerned. With overview to this manuscript, it seems a technique report, but not a real research article.  

1)       The title of manuscript is not appropriate since one main topic of the manuscript and lots statements in Introduction are concerned with PET cycling.

2)       After reading the whole manuscript, one may wonder what the main topic of the manuscript is. Is it “the synthesis of TMTS and its mechanism”, “the accelerator performance of TMTS in the vulcanization of rubber” or “recycling plasticizers from PET”? Thus, the parts of Abstract and Conclusion should be revised. The part of “the synthesis of TMTS and its mechanism” should be concise. Some description in Supplementary about other two parts is suggested to remove into the main manuscript.

3)       “plasticizers are generally used to reduce rubber relaxation time” in Line 62-63: the role of one plasticizers is to help the movability of polymeric chains, which can not reduce rubber relaxation time.

4)       aliphatic hydrophilic acids with high molecular weight” in Line 76: check if “hydrophilic” is right, considering most rubber is hydrophobic. The “high molecular weight” is suggested to be changed into “larger molecular weight” to avoid the misunderstanding.

5)       by oxidation of dithiocarbamates in high yield using NaHCO3” in Line 126: this expression is so surprised since NaHCO3 can not act as an oxidative reagent.

6)       The performance of TMTS as the curing accelerator in the literature is encouraged to offer in Introduction.

7)       Subsection 2.1 Materials: at least, the information of “Commercial rubber and additives” and chemical for synthesis of TMTS should be given therein.

8)       Line 187-188: what dose “to attain a negative reaction to SCN- ion” as well as “negative reaction to SCN- ion” in Line 208 suggest?

9)       5 cm3 (0.006 mol) of sodium methoxide” in Line 219: Is it the solution of sodium methoxide in methanol? What is the concentration?

10)   that the reaction takes place than when sulfuric acid is used” is suggested revised as “so that the reaction takes place compared with the usage of sulfuric acid”. Is this changed suitable?

11)   Line 415-416: it is difficulty to follow the statement, “By reducing the amount of water , to increase the productivity of the reactor”.

12)   Line 417-418: the statement, “which reduced the probability of adequately oriented collisions of the reacting particles”, is not needed.

13)   Line 488-489: it is not needed to “assume that the basicity of the investigated nucleophiles is correlated with the nucleophilicity of ”.

14)   J- (in superscript )” in L480, L491 and Scheme 6: it should be “I- (in superscript)”, right?

15)   Line 627-628: “The reaction took place so that the entire amount of potassium cyanide was added for 2 hours” is not good considering its expression.

16)   Tabl2 8 and 9: offer necessary explanation to “Mooney ML (1+4), 100 ºC”.

17)   Line 705: is it proper to revise “an oven for 1–5 days and provided with forced air circulation” as “an oven provided with forced air circulation for 1–5 days”?

18)   Line 754-755: please clarify the role of “amplifying materials”. Are they referred to strengthening additives for rubber or others?

19)   Conclusion should be re-written in concise way.

20)   Delete Line 900-924.

21)   the compound good resistance to aging” (L75) → “the obtained tires (or “the cured rubber”) good resistance to aging”; “then reacts” (L85) → “the intermediate product reacts”; “phase” (L91, L92, L181) → “step”; “in a ratio of 10:1” (L106) → “at a molar ratio of 10:1”; “and with stirring and heating at 55 0C” (L) → “under the condition of stirring and heating at 55 ºC”; “The yield dependence of the reaction product yield” (L403-404) → “The yield dependence of the reaction product disclosed”; “109 0C” (L440) → “o” is in superscript; “leading to the conclusion that the yellow crystals represented” (L441) → “confirms that the yellow crystals represents”; “was mixed” (L629) → “was stirred”; “thus improving” (L678) → “thus improves”.

The following comments are given to Supplementary.

22)   Subsection 2.4.1: “(10 mlsolution/g(Al2O3)” → “[10 mLsolution/g(Al2O3)]”. Offer the volume of “water solution”.

23)   Subsection 2.4.3: what does “150 o/min” mean?

24)   The subsection of Procedure, Page 9: what do “the first adsorber” and “the second adsorber” mean? Offer the concentrations of “solution of lead acetate” and “solution of potassium hydroxide”. Does “factor of iodine solution” refer to “concentration of iodine solution”?

25)   Analysis of moisture content, Page 9: it is so strange that the material is “dried on balance”.

26)   The writing of Supplementary is too tedious.

Author Response

Response to Reviewer 2 Comments

Thank you very much for your letter concerning our paper (processes-2246146) entitled: "One-pot synthesis of PET-based plasticizer and tetramethyl thiuram monosulfide (TMTS) as vulcanization accelerator for rubber production". The title was changed according to Reviewer 2 comment to be more appropriate for the results presented in manuscript.

We appreciate the opportunity to revise our work to be considered for publication in Processes. We have carefully taken the comment of the reviewers into consideration in preparing our revision. Our detailed reply to Reviewers' comments is given below and corrections were marked in yellow. Moreover, all the corrections of grammar and results obtained after checking were done in revised manuscript.

Reviewer 2

This manuscript comprises three topics, as mentioned in Comment 2. Although the part about the synthesis of TMTS and its mechanism is fully studied, others are relatively concerned. With overview to this manuscript, it seems a technique report, but not a real research article. 

Point 1: The title of manuscript is not appropriate since one main topic of the manuscript and lots statements in Introduction are concerned with PET cycling.

Response 1: The title "New process windows for the one-pot synthesis of tetramethyl thiuram monosulfide (TMTS) and its use as a rubber vulcanization accelerator"

was changed to:

"One-pot synthesis of PET-based plasticizer and tetramethyl thiuram monosulfide (TMTS) as vulcanization accelerator for rubber production", which is more appropriate for the results presented in manuscript.

Point 2: After reading the whole manuscript, one may wonder what the main topic of the manuscript is. Is it "the synthesis of TMTS and its mechanism", "the accelerator performance of TMTS in the vulcanization of rubber" or "recycling plasticizers from PET"? Thus, the parts of Abstract and Conclusion should be revised. The part of "the synthesis of TMTS and its mechanism" should be concise. Some description in Supplementary about other two parts is suggested to remove into the main manuscript.

Response 2: Abstract and Conclusion were rewritten.

Point 3:    "plasticizers are generally used … to reduce rubber relaxation time" in Line 62-63: the role of one plasticizers is to help the movability of polymeric chains, which can not reduce rubber relaxation time.

Response 3: The sentence "plasticizers are generally used to enhance rubber processability and to reduce rubber relaxation time in order to improve its physico-mechanical characteristics" is changed to "plasticizers are generally used to help the movability of polymeric chains and enhance rubber processability and to improve its physico-mechanical characteristics."

Point 4:    "aliphatic hydrophilic acids with high molecular weight" in Line 76: check if "hydrophilic" is right, considering most rubber is hydrophobic. The "high molecular weight" is suggested to be changed into "larger molecular weight" to avoid the misunderstanding.

Response 4: The part of the sentence: "aliphatic hydrophilic acids with high molecular weight" is replaced by aliphatic hydrophobic acids with larger molecular weight.

Point 5:       "by oxidation of dithiocarbamates in high yield using NaHCO3" in Line 126: this expression is so surprised since NaHCO3 can not act as an oxidative reagent.

Response 5: Oxidation of dithiocarbamates takes place but NaHCO3 is used for pH adjustment, not as an oxidation reagent. The correction has been made.

Point 6:       The performance of TMTS as the curing accelerator in the literature is encouraged to offer in Introduction.

Response 6: Literature review of TMTS as the curing accelerator was added to the Introduction section.

Point 7:       Subsection 2.1 Materials: at least, the information of "Commercial rubber and additives" and chemical for synthesis of TMTS should be given therein.

Response 7: All material data were transferred from Supplementary Material to the main document.

Point 8:       Line 187-188: what dose "to attain a negative reaction to SCN- ion" as well as "negative reaction to SCN- ion" in Line 208 suggest?

Response 8: The ammount of water was added: 200 cm3 of water (to attain a negative reaction to SCN- ion)

Point 9:    "5 cm3 (0.006 mol) of sodium methoxide" in Line 219: Is it the solution of sodium methoxide in methanol? What is the concentration?

Response 9: Yes, the solution of sodium methoxide is in methanol as a solvent and it is clearly designated in the text, as well the concentration: "in 5 cm3 (0.006 mol) of sodium methoxide solution (1.2 moldm-3) in dry methanol was added".

Point 10:   "that the reaction takes place … than when sulfuric acid is used" is suggested revised as "so that the reaction takes place … compared with the usage of sulfuric acid". Is this changed suitable?

Response 10: The sentence was rewritten, i.e. "that the reaction takes place … than when sulfuric acid is used" is replaced by "so that the reaction takes place … compared with the usage of sulfuric acid"

Point 11:   Line 415-416: it is difficulty to follow the statement, "By reducing the amount of water …, to increase the productivity of the reactor".

Response 11: The sentence is rewritten.

Point 12:   Line 417-418: the statement, "which reduced the probability of adequately oriented collisions of the reacting particles", is not needed.

Response 12: According to the suggestion, this part was deleted.

Point 13:   Line 488-489: it is not needed to "assume that the basicity of the investigated nucleophiles is correlated with the nucleophilicity of …".

Response 13: The sentence was rewritten: "Since the basicity of the investigated nucleophiles is correlated with the nucleophilicity of substitution on the S-atom in the S-S bond of TMTD, cyanide and amide ions used in these reactions gave higher product yields ".

Point 14:  "J- (in superscript )" in L480, L491 and Scheme 6: it should be "I- (in superscript)", right?

Response 14: Right, the symbol J- is replaced with I-.

Point 15:   Line 627-628: "The reaction took place so that the entire amount of potassium cyanide was added for 2 hours" is not good considering its expression.

Response 15: The sentence was changed to "The entire amount of potassium cyanide was added for 2 hours".

Point 16:   Table2 8 and 9: offer necessary explanation to "Mooney ML (1+4), 100 ºC".

Response 16: An explanation of Mooney viscosity measurement was added in the main manuscript (Results section 3.6. Rubber products properties)

Point 17:   Line 705: is it proper to revise "an oven for 1–5 days and provided with forced air circulation" as "an oven provided with forced air circulation for 1–5 days"?

Response 17: The part of the sentence "an oven for 1–5 days and provided with forced air circulation" was replaced with "an oven provided with forced air circulation for 1–5 days."

Point 18:   Line 754-755: please clarify the role of "amplifying materials". Are they referred to strengthening additives for rubber or others?

Response 18: Yes, they referred to strengthening additives for rubber. The phrase "amplifying materials" was replaced with "reinforcing materials".

Point 19:   Conclusion should be re-written in concise way.

Response 19: The Conclusion was rewritten and shortened to be concise and more precise.

Point 20:   Delete Line 900-924.

Response 20: Lines 900-924 were deleted.

Point 21:   "the compound good resistance to aging" (L75) → "the obtained tires (or "the cured rubber") good resistance to aging"; "then reacts" (L85) → "the intermediate product reacts"; "phase" (L91, L92, L181) → "step"; "in a ratio of 10:1" (L106) → "at a molar ratio of 10:1"; "and with stirring and heating at 55 0C" (L) → "under the condition of stirring and heating at 55 ºC"; "The yield dependence of the reaction product yield" (L403-404) → "The yield dependence of the reaction product disclosed"; "109 0C" (L440) → "o" is in superscript; "leading to the conclusion that the yellow crystals represented" (L441) → "confirms that the yellow crystals represents"; "was mixed" (L629) → "was stirred"; "thus improving" (L678) → "thus improves".

Response 21: All suggestions were accepted. The following comments are given to Supplementary.

Point 22:   Subsection 2.4.1: “(10 mlsolution/g(Al2O3)” → “[10 mLsolution/g(Al2O3)]”. Offer the volume of "water solution".

Response 22: Changed.

Point 23:   Subsection 2.4.3: what does “150 o/min” mean?

Response 23: 150 o/min is replaced with 150 rpm

Point 24:   The subsection of Procedure, Page 9: what do "the first adsorber" and "the second adsorber" mean? Offer the concentrations of "solution of lead acetate" and "solution of potassium hydroxide". Does "factor of iodine solution" refer to "concentration of iodine solution"?

Response 24: Adsorbers are used for the bubbling of the gaseous product from the reactor outlet in order to obtain solution used for CS2 and H2S quantitative determination:

  • adsorber for hydrogen sulfide exhaust gas (the first adsorber) filled with 10% lead acetate, aqueous solution
  • adsorber for carbon disulfide exhaust gas (the second adsorber) filled with 2M methanol solution of potassium hydroxide, methanol solution.

The factor of iodine solution (F) is the number that describes the molar concentration of the reference substance used in volumetric analysis. To obtain more precise molar concentration of iodine solution, the so-called titer determination or standardization of a volumetric solution used for titration is one of the most important preconditions for reliable and transparent titration results. The accurate molar concentration is obtained by multiplying the initial C by factor F.

Point 25:   Analysis of moisture content, Page 9: it is so strange that the Material is "dried on balance".

Response 25: It is a mistake, the wrong description was corrected.

Point 26:   The writing of the Supplementary is too tedious.

Response 26: According to the reviewer's suggestion, we moved a part of the Supplementary Material to the main manuscript.

Round 2

Reviewer 1 Report

The authors have addressed the comments and queries. 

Author Response

The English language and grammar were carefully checked.

Reviewer 2 Report

In the previous reviewing comments, it was said that “with overview to this manuscript, it seems a technique report, but not a real research article”. Although this revised manuscript has been carefully revised according to the previous comments, the reviewer still has this opinion. Whether or not this manuscript will be finally accepted depends the judgment of the editors. The following is some suggestions and questions.  It is not needed to return the further revised manuscript to the reviewer. 

1) The title of revised manuscript suggested that “PET-based plasticizer” was prepared through one-pot synthesis and used as vulcanization accelerator. Is it right?

2) The previous study about the performance of TMTS as the curing accelerator in the literature is encouraged to offer and discuss in Introduction.

3) Subsection 2.1: in Line 124~125, there are two “sodium cyanide”s from two suppliers. Shorten this subsection and clearly offer the information about commercial rubbers and their additives.

4) Line 171, Line 190 and Line 571: was “negative reaction” referred to “side reaction” or “by-reaction”?

5) Line 441-442: “the basicity of the investigated nucleophiles” and “the nucleophilicity of substitution” is of the same meaning, which was mentioned in the previous comment.

6) Line 444-445: “the good overlap of the polarizable 5p-orbital with the 3p-orbital of sulfur” may be changed into “the good overlap of its polarizable 5p-orbital with the 3p-orbital of sulfur”.

7) Line 509-519: keep the verb times consistent, like “the reaction does not take place in one step but that intermediates were present”.

8) Line 584: the statement that “E factors up to around 500 may be acceptable” is strongly questionable since only 0.2 % of waste can be recycled based on the definition in Line 578. 

9) Line 618-619: besides “the rubber compound”, other components of filler, oil and additives also affect “processability”. 

10)  “Tetrabutyl Titanate” (L129) → “Tetrabutyl titanate”; “γ-Alumina” (L131) → “γ-alumina”; “1.2 moldm-3” (L200) → “1.2 mol/dm-3”, keeping the unit the same as that in Line 511, so does for Line 213, Line 604 and 604; “propylene glycol (PG) levulinic acid (LA)” (L294) → “propylene glycol (PG), levulinic acid (LA)”; “As a results” (L383) → “Thus, ”; “leading to the conclusion that the yellow crystals represented” (L404) → “confirms that the yellow crystals represents”; “The reaction is first order” (L513) → “The reaction is the first order”; “in more and close to first order with” (L514) → “while in more and close to the first order with”; “which directly relate to torque” (L639) → “which is directly related to torque”.

Author Response

Response to Reviewer 2 Comments

Thank you very much for your comments concerning our paper (processes-2246146) entitled: "One-pot syntheses of PET-based plasticizer and tetramethyl thiuram monosulfide (TMTS) as vulcanization accelerator for rubber production." The title was changed according to Reviewer 2 comment to be more appropriate for the results presented in the manuscript.

We again appreciate the opportunity to revise our work to be improved for publication in Processes. We have carefully considered the comment of the reviewers in preparing our revision. Our detailed reply to Reviewer's comments is given below, and corrections were made to the manuscript. The English language and grammar were also checked.

Point 1: The title of the revised manuscript suggested that the “PET-based plasticizer” was prepared through one-pot synthesis and used as a vulcanization accelerator. Is it right?

Response 1: The title is changed to “One-pot syntheses of PET-based plasticizer and tetramethyl thiuram monosulfide (TMTS) as vulcanization accelerator for rubber production”. The synthesis of PET-based plasticizer is the one, and the synthesis of TMTS is the other. Propylene glycol (PG) and levulinic acid (LA) were obtained from bioresources, and LA/EG/PET/PG/LA, obtained from waste PET, PG and LA, represent PET-based plasticizers, while TMTS is used as a vulcanization accelerator.

Point 2: The previous study about the performance of TMTS as the curing accelerator in the literature is encouraged to offer and discuss in Introduction.

Response 2: Though TMTS is well-known as a vulcanization accelerator, few research studies are related to its performance. We added some references about its use in rubber production in the Introduction.

Point 3: Subsection 2.1: in Line 124~125, there are two “sodium cyanide”s from two suppliers. Shorten this subsection and clearly offer the information about commercial rubbers and their additives.

Response 3: Subsection 2.1. was shortened, and we added all the information about commercial rubbers and additives.

Point 4: Line 171, Line 190 and Line 571: was “negative reaction” referred to “side reaction” or “by-reaction”?

Response 4: No, it is neither side reaction nor by-reaction. „Negative reaction to SCN- ion“refers to a „negative SCN- test, “i.e., the test reaction that shows no more SCN- ions in the solution, that they are completely removed by washing with water. However, to clarify, „Negative reaction to SCN- ion“ is changed to „negative SCN- test.“ This test is performed by a colorimetric method for the determination of thiocyanate ion.

Point 5: Line 441-442: “the basicity of the investigated nucleophiles” and “the nucleophilicity of substitution” is of the same meaning, which was mentioned in the previous comment.

Response 5: Though basicity and nucleophilicity are similar concepts, they are not the same. Basicity is the ability to accept hydrogen, but nucleophilicity is the ability to attack substrate/molecule to initiate a certain reaction. All nucleophiles are bases, but all bases cannot be nucleophiles. On the other side, whereas nucleophilicity considers the reactivity (i.e., the rate of reaction) and represents the kinetic term, basicity is a measure of the equilibrium position (thermodynamic term). However, nucleophilicity correlates with basicity, as we said in that part of the manuscript: “Since the basicity of the investigated nucleophiles is correlated with the nucleophilicity of nucleophiles in the substitution reaction on the S-atom of the S-S bond of TMTD, cyanide and amide ions used in these reactions gave higher product yields...“

Point 6: Line 444-445: “the good overlap of the polarizable 5p-orbital with the 3p-orbital of sulfur” may be changed into “the good overlap of its polarizable 5p-orbital with the 3p-orbital of sulfur”.

Response 6: Changes are made according to reviewer's suggestion.

Point 7: Line 509-519: keep the verb times consistent, like “the reaction does not take place in one step but that intermediates were present.”

Response 7: The sentence “the reaction does not take place in one step, but that intermediates were present” is changed to “the reaction does not take place in one step, but that intermediates are present.”

Point 8: Line 584: the statement that “E factors up to around 500 may be acceptable” is strongly questionable since only 0.2 % of waste can be recycled based on the definition in Line 578. 

Response 8: The detailed explanation of the E factor we found in the book by Stanley E. Manahan -Green Chemistry and the Ten Commandments of Sustainability:

In Chapter 13.10: The E-factor in Green Chemistry, it is written:

“E factors up to around 500 may be acceptable if the value of the
product is high enough to justify the cost of treating and disposal of wastes.”

 That reference also states: “In the pharmaceutical manufacturing industry where annual quantities generated typically are measured in tens to several hundred tons per year, acceptable E factors may be up to 4000.” However, we also added this reference in the manuscript.

Point 9: Line 618-619: besides “the rubber compound”, other components of filler, oil and additives also affect “processability”. 

Response 9: Of course, other components also affect processability. We didn’t mean that only the rubber compound affects the processability. Still, to better understand, we changed this sentence to:” It depends on the cure characteristics and rheological properties of the rubber compound, as well as other components of filler, such as oil and additives.”

Point 10:   “Tetrabutyl Titanate” (L129) → “Tetrabutyl titanate”; “γ-Alumina” (L131) → “γ-alumina”; “1.2 moldm-3” (L200) → “1.2 mol/dm-3”, keeping the unit the same as that in Line 511, so does for Line 213, Line 604 and 604; “propylene glycol (PG) levulinic acid (LA)” (L294) → “propylene glycol (PG), levulinic acid (LA)”; “As a results” (L383) → “Thus, ”; “leading to the conclusion that the yellow crystals represented” (L404) → “confirms that the yellow crystals represents”; “The reaction is first order” (L513) → “The reaction is the first order”; “in more and close to first order with” (L514) → “while in more and close to the first order with”; “which directly relate to torque” (L639) → “which is directly related to torque”.

Response 10: All suggestions were accepted and included in the manuscript.